# Simultaneous Screening of 322 Residual Pesticides in Fruits and Vegetables Using GC-MS/MS and Deterministic Health Risk Assessments

**DOI:** 10.3390/foods12163001

**Published:** 2023-08-09

**Authors:** Byong-Sun Choi, Dong-Uk Lee, Woo-Seong Kim, Chan-Woong Park, Won-Jo Choe, Myung-Jun Moon

**Affiliations:** 1Department of Industrial Chemistry, Pukyong National University, Busan 48513, Republic of Korea; sunny2597@naver.com; 2Biomedical Manufacturing Technology Center, Korea Institute of Industrial Technology, Yeongcheon 38822, Republic of Korea; young2home@pukyong.ac.kr; 3Center of Food & Drug Analysis, Busan Regional Office of Food and Drug Safety, Ministry of Food and Drug Safety, Busan 47537, Republic of Korea; kwsh1964@korea.kr (W.-S.K.); pcw0324@korea.kr (C.-W.P.); 4Pesticides & Veterinary Drug Residues Division, National Institute of Food & Drug Safety Evaluation, Ministry of Food and Drug Safety, Cheongju 28159, Republic of Korea

**Keywords:** multiresidue pesticide analysis, food safety, QuEChERS, agricultural product, mass spectrometry

## Abstract

The development of efficient methods for evaluating pesticide residues is essential in order to ensure the safety and quality of agricultural products since the Republic of Korea implemented the Positive List System (PLS). The objective of this research was to establish a method for the simultaneous analysis of 322 pesticide residues in fruits and vegetables (such as coffee, potato, corn, and chili pepper), using the quick, easy, cheap, effective, rugged, and safe (QuEChERS) approach in combination with gas chromatography-tandem mass spectrometry (GC-MS/MS). This study introduces a robust, high-throughput GC-MS/MS method for screening the target pesticide residues in agricultural products, achieving the PLS criterion of 0.01 mg/kg LOQ. Despite some compounds not aligning with the CODEX recovery guideline, sufficient reproducibility was confirmed, attesting to the method’s applicability in qualitative analyses. A health risk assessment conducted using estimated daily intake/acceptable daily intake ratios indicated low risks associated with product consumption (<0.035391%), thereby confirming their safety. This efficient method holds significant implications for the safe distribution of agricultural products, including during import inspections.

## 1. Introduction

Pesticides are essential substances in agriculture that protect crops against harmful insects to maximize yields and enhance the quality of agricultural products. Despite the apparent benefits of pesticides, their indiscriminate use causes problems because residues in agricultural products can have adverse effects on human health and cause environmental pollution [1,2,3]. The agricultural products in each country are cultivated according to the individual conditions in that country, and the type of pesticide used varies according to the environment and the type of pest emerging during the cultivation period [4,5]. Hence, each country defines their own maximum residue limits (MRLs) for agricultural products to ensure efficient and rigorous pesticide management and control, to guarantee the safety of domestic and imported agricultural products, and to conduct continuous monitoring of the agricultural products being distributed [6,7]. In addition, consumers often base decisions regarding the purchase of agricultural products on health and safety concerns; therefore, restricting the use of pesticides on agricultural products and analyzing for pesticide residues are important.

Pesticide residue analyses are broadly classified into individual methods and multiclass multiresidue methods (MRMs) based on the purpose of the analysis [8,9]. Individual methods are highly reliable because the analysis is optimized for individual compounds; however, these methods are less frequently applied for pesticide residue analysis, owing to the time and cost required for detecting multiple pesticides [10,11]. In contrast, MRMs aim to analyze different classes of pesticides in a single analysis, which is conducive for the rapid processing of a large number of agricultural products [8,12]. Based on this efficiency, MRMs are mainly used in the routine monitoring of various pesticide residues and are widely used worldwide for the safety management of pesticide residues [13,14].

In the field of multi-pesticide residue analysis, gas chromatography-mass spectrometry (GC-MS) and liquid chromatography (LC) techniques are acceptable. Especially, GC-MS or GC-tandem MS (GC-MS/MS) offers high sensitivity and selectivity, a broad analytical spectrum, quantitative accuracy, exceptional resolution, and fast measurement speed, as well as high-quality identification of various substances. However, a highly suitable sample preparation approach such as quick, easy, cheap, effective, rugged, and safe (QuEChERS) has been recognized and is required before instrumental analysis [15,16,17]. In contrast to the long extraction times required when using organic solvents to extract pesticides from different samples, the analysis time and processing steps are minimized in QuEChERS, which still exhibits high recoveries of 60–120% for various nonpolar and polar pesticides [18,19,20]. Since its adoption as an Association of Official Analytical Chemists (AOAC) method in 2003, the recently improved QuEChERS method has been approved as an official method of analysis in a number of European countries (EN 15662 method) (Figure 1) [21].

Several previous studies have developed MRMs using the QuEChERS method to analyze various pesticides [22,23,24]; however, for such MRMs to be utilized in the Republic of Korea, these methods must be validated for foods frequently consumed by Koreans [25,26,27,28,29]. In 2019, the government of the Republic of Korea introduced the Positive List System (PLS), which applies a uniform MRL criterion (0.01 mg/kg) to all imported agricultural products for which a specific MRL has not been established in Korea. Moreover, mandatory testing of pesticide residues on imported agricultural products has been instituted. Nevertheless, numerous pesticides still lack established analytical standards from the Korean Ministry of Food and Drug Safety (Figure 2 and Appendix A).

This study undertook the challenge of simultaneously screening 322 pesticides (359 compounds in total, including isomers and metabolites) that are predominantly used in Korean agriculture. The screening was conducted using GC-MS/MS and validated on four types of agricultural products that are heavily imported and frequently consumed in the Republic of Korea: coffee, potato, corn, and chili pepper [28,29,30,31]. Given the dietary habits of Koreans, these agricultural products are particularly suitable for a study aimed at minimizing the health risks posed by residual pesticides. The Limit of Quantification (LOQ) of the method satisfied the PLS criterion of 0.01 mg/kg, and the linearity, accuracy, and precision of the method were verified. Therefore, the findings of this study could significantly contribute to the development of screening standards and safety management, and aid in food safety management in Korea.

## 2. Materials and Methods

### 2.1. Chemicals and Reagents

The standard materials of 322 pesticides (359 compounds in total, including isomers and metabolites) used in the analysis were purchased from AccuStandard (New Haven, CT, USA), Chemservice (West Chester, PA, USA), Dr. Ehrenstorfer (Augsburg, Germany), Wako (Osaka, Japan), Fluka (Udligenswil, Switzerland), Sigma-Aldrich (St. Louis, MO, USA), and Supelco (Bellefonte, PA, USA). The acetonitrile (ACN) and acetone used in the extraction and purification processes were high-purity GC- or pesticide residue-grade solvents purchased from Merck (Darmstadt, Germany). The solid-phase extraction (SPE) kit, used in this study, was purchased from Applied Separation (Hamilton, PA, USA). Polytetrafluoroethylene membrane filters (PTFE, 0.2 μm) and EN QuEChERS salts (QuEChERS EN 15662 Method Extraction Kits) were purchased from Thermo Fisher Scientific (Waltham, MA, USA).

### 2.2. Standard Solutions

The 322 standard pesticide solutions (1000 mg/L) were separately prepared in 20 mL of acetone. The commercial standard solutions were also purchased and kept at −18 °C until use. The working standard solution comprised a mixture of each standard stock solution diluted to a set certain concentration with acetone in a brown bottle and stored at 4 °C. The working solution was diluted before use in each analysis.

### 2.3. Analysis by GC–Triple Quadrupole MS/MS

The GC system used for the simultaneous analysis of the pesticide residues was a GC-2010 (Shimadzu, Japan) with a split/splitless injector (SSI) and MS/MS systems (TQ8040, Shimadzu, Japan). The column used to separate the pesticide compounds was a DB-5MS (30 m × 0.25 mm × 0.25 μm, Agilent, Santa Clara, CA, USA). The GC was operated in splitless mode to analyze the 322 pesticide compounds in a 2 μL injection. To ensure the efficient separation of the compounds on the GC column, the initial oven temperature was set at 70 °C, which was maintained for 1 min at 0.8 mL/min of the He carrier gas. The temperature was then increased at a constant rate to a final temperature of 300 °C, which enabled the separation of 322 pesticides within 40 min. The detailed measurement conditions are shown in Table 1.

After separating the compounds on the capillary column, they were ionized using positive mode electron ionization (EI) at 70 eV, which is commonly used in pesticide residue analysis. The conditions were as follows: source temperature, 200 °C; transfer line temperature, 250 °C; manifold temperature, 40 °C; and detector voltage, 1400 V. The compounds were detected using the multiple reaction monitoring (MRM) mode, with a solvent delay of 3 min to protect the detector by allowing the solvent to pass through. The He carrier gas and Ar collision cell gas both had a purity of ≥99.999%.

Table 2 presents the optimal MRM conditions for the 322 pesticide compounds (Appendix A). To establish the optimal MRM analysis conditions, the 322 pesticides were thoroughly mixed and prepared to a 5 mg/L working standard solution. The solution was injected into the device in full scan mode (Figure 3), and the total ion chromatogram (TIC), mass spectrum, and retention time were obtained for each pesticide. Since the molecular masses of all 322 pesticide compounds were ≤500, the full scan range was set at 50–500 *m/z*, and each spectrum was compared with that of known pesticide compounds in the NIST library to confirm the similarity and verify an accuracy of the qualitative information for each pesticide.

From the full scan spectrum, a representative ion was selected as the precursor ion based on the molecular structure and unique mass fragments of each pesticide compound. The precursor ions that passed through the first quadrupole analyzer were reionized at 5–50 eV in the collision cell, and the resulting product ions passed through the second quadrupole analyzer to obtain the ion chromatogram of the individual mass fragments (Figure 3). At least two product ions were selected for each pesticide compound by comparing and considering the peak shapes and sensitivities, and then selecting suitable ion signals and collision energies. The confirmation criteria for the pesticide compounds were as follows: two or more product ions had to be detected and the height ratio of each ion had to match that in the reference spectrum. For quantification, the chromatographic area ratio of the product ions was compared to that in the standard.

In quadrupole MS analyses, several ions can simultaneously reach the detector within a short time to produce ambiguous peak shapes, resulting in reduced sensitivity. To prevent this problem, the dwell time of the MRM ions was set within a range of 0.15–0.29 s and the scan speed was set to 0.2–0.82 s/scan.

### 2.4. Sample Preparation and Extraction

Figure 4 shows a sample preparation process. After homogenizing the samples using a large-volume grinder, 10 g of chili pepper and potato samples and 5 g of low-water-content coffee and corn samples (purchased from local supermarkets) were accurately weighed. The low-water-content samples were mixed with 10 mL of distilled water and the mixtures were left to stand for 30 min for moistening before extraction. For pretreatment, the EN15662 QuEChERS method, including pH control, was applied [32]. The prepared sample was placed in a 50 mL polypropylene centrifuge tube to which 10 mL of ACN was added as an organic solvent. The sample underwent vigorous vortex mixing for 1 min to ensure that the solvent and the sample were adequately mixed. The EN kit (MgSO_4_, 4 g; NaCl, 1 g; sodium hydrogen citrate sesquihydrate, 0.5 g; and sodium citrate dihydrate, 1 g) was added to the tube and after 1 min of vigorous shaking, the mixture was centrifuged for 10 min (4000× *g* at 4 °C). During this process, the water and organic solvent (ACN) layers clearly separated and the pesticide compounds remaining in the sample were mostly transferred to the organic solvent layer. The pesticides and the water in the sample mixed with the ACN, and the powerful dehydration capabilities of the MgSO_4_ and NaCl forcefully separated the water from the organic solvent. The centrifugation process considerably reduced the time required to separate the water and organic solvent layers.

To a 2 mL centrifuge tube containing 150 mg of MgSO_4_ and 25 mg of a primary secondary amine (PSA) sorbent, 1 mL of the supernatant obtained in the extraction process was added and the lid was closed for 1 min of vigorous shaking. Next, the mixture was centrifuged for 15 min (4000× *g* at 4 °C) to ensure the adequate separation of the layers. The supernatant was passed through a membrane filter (PTFE, 0.2 μm), and the final extract was transferred to a brown vial for analysis [32].

### 2.5. Method Validation

The method for the simultaneous analysis of 322 pesticides was validated according to the CODEX guidelines for pesticide residue analysis (CAC/GL40) and the Guideline of Standard Procedures of Test Methods for Foods and Other Substances (April, 2016) published by the Ministry of Food and Drug Safety (MFDS, Republic of Korea) [33]. The method developed in this study was validated for linearity, LOQ, recovery, and reproducibility.

The linearity was determined via serial dilution of a working standard solution containing a mixture of the 322 pesticides to 5, 10, 50, 100, and 200 ng mL^−1^ to obtain a calibration curve for each compound. The LOQ was determined by diluting the working standard solution with purified extracts of the coffee, potato, corn, and chili pepper samples, which were selected as representative agricultural products for the validation. On the resulting chromatograms, the concentration of the standard solution with a signal-to-noise ratio (S/N) of 10 was determined and verified [34]. For the recovery and reproducibility, the coffee, potato, corn, and chili pepper samples were spiked with various concentrations of the working standard solution (0.005, 0.01, 0.05, 0.1, and 0.2 mg/L) and left to stand for 30 min to allow the pesticide compounds to adequately mix with the sample. The spiked samples were then extracted and analyzed using the method developed in this study. The percentage recovery (%) was determined by comparing the experimentally determined concentration with the added amount of pesticide. To verify the reproducibility, the relative standard deviation (RSD) of the recoveries was calculated from five repeated measurements per concentration.

Matrix Effects (ME) are defined by the CODEX as influences on the measured concentration or the amount of the target analyte due to other components within the sample. The extent of the ME, determined by contrasting the response of the analyte in a pure standard solution with its response in a sample extract, can differ significantly (Appendix A). The ME is evaluated by comparing the slope of the calibration curves for the standards in the solvents against the standards prepared in the ME. The ME is calculated using Equation (1): [35].
(1)$$\mathrm{Matrix}\;\mathrm{effects}\;\left(\mathrm{ME}\right)=\left(\frac{\mathrm{Slope}\;\mathrm{of}\;\mathrm{calibration}\;\mathrm{curve}\;\mathrm{in}\;\mathrm{matrix}}{\mathrm{Slope}\;\mathrm{of}\;\mathrm{calibration}\;\mathrm{curve}\;\mathrm{in}\;\mathrm{solvent}}-1\right)\times 100\%$$

If the ME exhibits a suppression or enhancement of 0–20%, it is referred to as a Soft Matrix Effect, which is generally negligible and does not significantly impact the analysis. However, if the suppression or enhancement is between 20 and 50%, the ME is considered Medium, which may influence the analysis. In cases where the suppression or enhancement exceeds 50%, the ME is classified as Strong [36]. In such instances, it is necessary to implement measures to mitigate the ME’s impact on the analysis. One recommended approach is to use matrix-matched calibration, which involves the preparation of calibration standards in untreated or non-detect samples following the same pretreatment process. This helps in reducing the analytical error due to the ME when the sample extract and matrix-matched standard are injected into the instrument for analysis. Another potential approach to tackle a Strong ME could be sample dilution. The ME % values are presented in Appendix A.

### 2.6. Monitoring Using the Multiresidue Method for Pesticide Residue Analysis

The field applicability of the MRMs developed in this study was verified by determining the current state of pesticide residues in selected agricultural products distributed in the Republic of Korea. Samples were collected from three cities in the Republic of Korea: Busan, Ulsan, and Gimhae. The agricultural products selected for the monitoring were those commonly available at markets and frequently consumed by Koreans [24]. A total of 135 samples were purchased and tested. To collect accurate data and assess the pesticide exposure of the national population from the consumption of agricultural products, the selected samples were agricultural products from the final stage before consumption, such as those available at major supermarkets or wholesale markets, rather than those from the production stage, because the goal was to examine agricultural products right after purchase and before consumption. Table 3 presents the type and number of the sampled agricultural products.

Based on the type of agricultural product, 1 kg of fresh product or 0.3 kg of dried product was purchased to ensure that a representative amount of each sample was collected, as per the Korean Food Code requirements [24]. The collected samples were immediately transferred to the laboratory and the entire amount was homogenized in a large-volume grinder and pretreated according to the method developed in this study. Excess samples remaining after the experiments were divided among the sealed containers and stored in a freezer (−18 °C) to prevent the partial degradation of the pesticides by light or temperature in case reanalysis was required.

## 3. Results and Discussion

### 3.1. Analytical Method Validation

#### 3.1.1. Linearity

To determine the linearity of the 322 pesticide compounds, the working standard solutions were prepared at five concentrations (0.005, 0.01, 0.05, 0.1, and 0.2 mg/L) in the LOQ range of 0.005–0.2 mg/L and injected into the device for analysis. The coefficient of determination (*R*^2^) was ≥0.98 in all cases, indicating a high level of linearity and confirming that the method was suitable for quantitative analyses. Appendix A presents the correlation coefficients for the four agricultural products.

#### 3.1.2. Recovery

To determine the recovery, the analysis was repeated five times at each spiking level for each target sample. The number of pesticides within the mean recovery range of 60–120% was 349 out of 359 for coffee, 348 for potato, 339 for corn, and 346 for chili pepper. The standard deviation (SD) of the repeated recovery tests was mostly within 30%, which satisfied the CODEX requirement for pesticide residue analysis (Table 4 and Appendix A). The primary aim of this study was the development of a high-throughput screening tool capable of rapidly detecting a broad spectrum of pesticides. Achieving an RSD of 15% or less is indeed desirable, yet not consistently feasible given the intrinsic variability of the matrices and the multitude of target compounds. Despite certain constraints, this methodology, with its capacity for swift screening of a wide array of pesticides across diverse commodities, carries significant practical implications. Thus, it needs refinement and optimization with a specific focus on commodities such as coffee and chili pepper in subsequent research endeavors.

#### 3.1.3. Limit of Quantification

Among the target pesticides in this study, certain compounds did not have an established MRL for specific agricultural products. Thus, to develop a method that can detect pesticides at a level of ≤0.01 mg/kg, which is the PLS criterion of non-detection required for pesticides, 0.01 mg/kg extracts of coffee, potato, corn, and chili pepper samples were prepared using the pretreatment method developed in this study. A S/N ratio of ≥10 was obtained for all 322 pesticides in the four samples. This confirms that the developed method is suitable for determining if a given agricultural product is in compliance with the ≤0.01 mg/kg MRL required by the PLS. Appendix A shows chromatograms for GC-MS/MS pesticides.

Based on the results, the MRM developed in this study was confirmed to comply with the international standard and enables the simultaneous qualitative and quantitative analyses of pesticide compounds using a single pretreatment process. Certain compounds were not within the guideline recovery range; however, a certain level of reproducibility was maintained at a low LOQ of ≤0.01 mg/kg. This indicates that, although the method may not be suitable for the quantitative analyses of these compounds, it can be used for qualitative analyses to confirm the presence of pesticide residues in food products.

### 3.2. Pesticide Residue Concentration in Agricultural Products

To investigate the current state of pesticide residues in commercially available agricultural products, ten products that are frequently consumed in high amounts (chili pepper, carrot, garlic stem, mango, wheat, banana, almond, cabbage, coffee, and pineapple) were purchased from major supermarkets or wholesale markets in regions of the Republic of Korea, including Busan, Gimhae, and Yangsan. The amount of each product that was purchased was determined based on the level of consumption by the national population. Consequently, the highest number of samples taken was for bananas (24 samples) and the lowest number was for garlic stems (seven samples). A total of 135 samples of agricultural products were collected; the number of samples per agricultural product and the detection results are shown in Table 5.

In the 135 samples of ten types of agricultural products, three pesticides were detected: chlorpyrifos, fludioxonil, and prochloraz, and an exposure assessment was conducted. The daily intake was estimated based on the amount of pesticide detected and the corresponding consumption of the agricultural product. Table 6 shows the estimated daily intake (EDI) against the acceptable daily intake (ADI) (EDI/ADI %), which is calculated using the 7th Korea National Health and Nutrition Examination Survey (KNHANES VII-1). Considering Korean dietary habits, the EDI of pesticides was calculated according to pesticide intake amount (mg) per kg of body weight.

The exposure assessments on the three pesticides, chlorpyrifos, fludioxonil, and prochloraz, revealed that the values of the EDI against the ADI (EDI/ADI) were 0.002995%, 0.000187% (Mango), 0.000062% (Pineapple), and 0.035391%, respectively. This indicated that the health risk from consuming the residual pesticides on the collected agricultural products was considerably low and would further decrease during processes such as washing and cooking [37,38].

## 4. Conclusions

The proposed GC-MS/MS, combined with the QuEChERS method, has been successfully employed for the simultaneous multi-pesticide residue analysis in Korean agricultural products. This methodology demonstrated high selectivity and sensitivity with satisfying the PLS criterion of 0.01 mg/kg for the LOQ. Moreover, the EDI and ADI were also calculated to facilitate the assessment of potential health risks (EDI/ADI) posed by the analyzed products. The findings of this research establish the comprehensive screening standard method for various pesticides and advances food safety management. Further studies expanding to other types of pesticides in various agricultural products will be essential for improving food safety measurement and public health in Korea.

## Figures and Tables

**Figure 1 foods-12-03001-f001:**
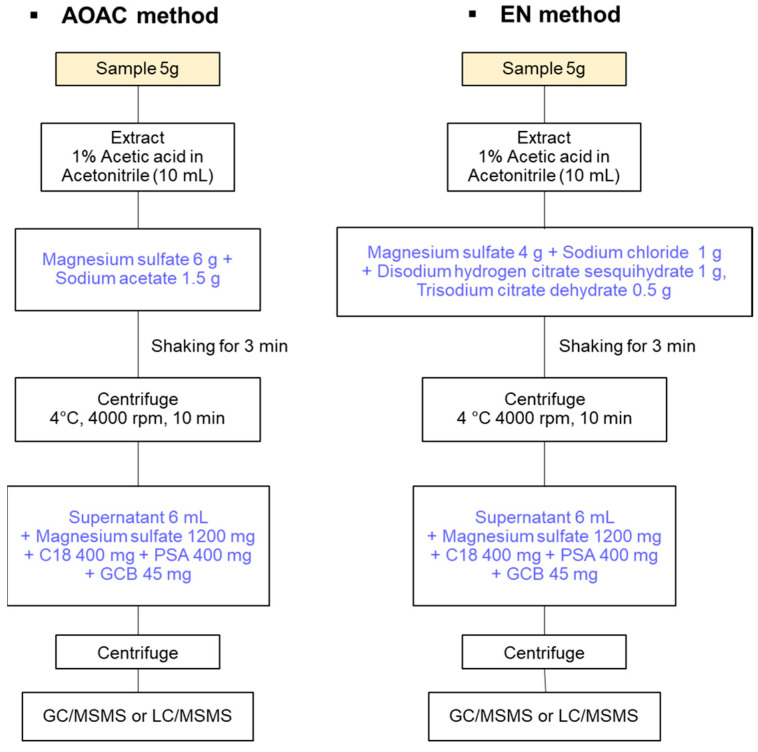
Procedure of types of QuEChERS methods.

**Figure 2 foods-12-03001-f002:**
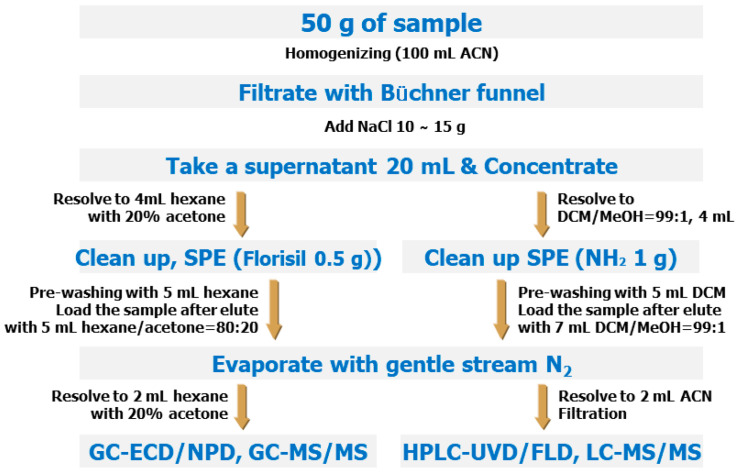
Analytical method for MRMs of the Korean Food Standards Codex.

**Figure 3 foods-12-03001-f003:**
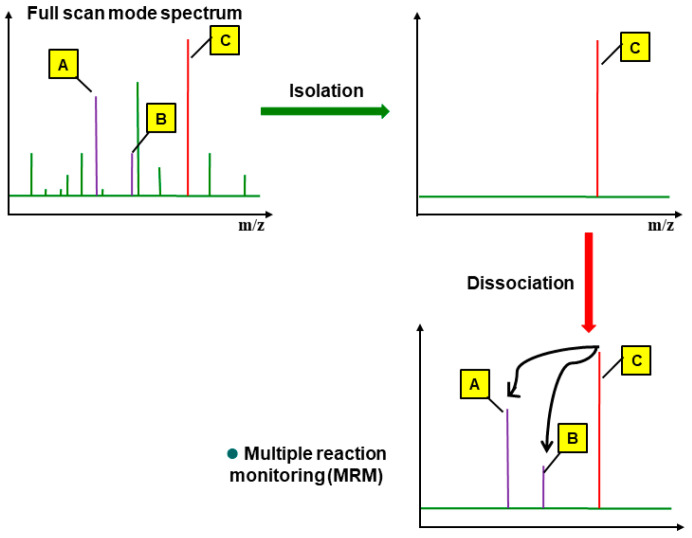
Procedure of multiple reaction monitoring (MRM). A, B, and C are representing qualitative ion 2, qualitative ion 1, and quantitative ion, respectively.

**Figure 4 foods-12-03001-f004:**
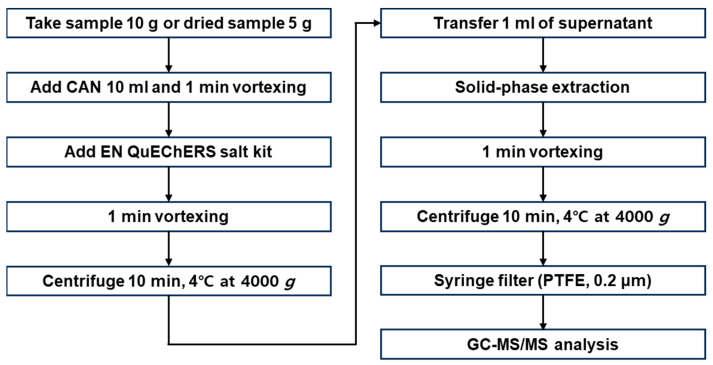
Schematic diagram of sample preparation.

**Table 1 foods-12-03001-t001:** Instrumental conditions for the analyses of pesticides.

Instrument	GC-2010 GC body with AOC-20i plus autoinjector (Shimadzu, Japan)			
Injection vol., mode	2 μL, split/splitless			
Column	HP-5 ms, 30 m × 0.25 mm i.d., 0.25 μm (Agilent US)			
Carrier Gas	Helium (0.8 mL/min)			
Column oven	Rate (°C/min)	Temp (°C)	Hold (min)	Total (min)
	Initial	70	3	3
	20	180	0	8.5
	5	300	7.5	40
Scan	MRM mode			
Source Temp.: 200 °C; Detector Temp. (transfer line): 250 °C				

**Table 2 foods-12-03001-t002:** GC-MS/MS parameters for the analysis of 322 pesticides.

No.	Pesticides	RT (min)	MRM Transitions					
			Quantitative Ion	CE (eV)	Qualitative Ion 1	CE (eV)	Qualitative Ion 2	CE (eV)
1	2,6-Diisoporpylnaphthalene	9.665	197 > 155	12	212 > 197	15	212 > 155	24
2	Acetochlor	12.088	223 > 147	10	223 > 132	20	146 > 131	20
3	Acibenzola_s_methyl	12.318	182 > 152	30	182 > 135	20	182 > 107	35
4	Acrinathrin_1	23.502	208 > 181	5	208 > 152	30	181 > 152	40
	Acrinathrin_2	23.93	208 > 181	5	208 > 152	30	181 > 152	40
5	Alachlor	12.402	188 > 160	10	188 > 132	20	188 > 131	20
6	Aldrin	13.491	263 > 193	33	263 > 191	39	293 > 186	39
7	Allethrin-1	15.017	123 > 81	9	123 > 79	15	136 > 93	14
	Allethrin-2	15.161	123 > 81	9	123 > 79	15	136 > 93	14
8	Allidochlor	5.087	138 > 96	6	96 > 56	9	138 > 81	9
9	Ametryn	12.405	227 > 170	12	227 > 185	6	227 > 58	14
10	Anilofos	22.092	226 > 157	15	226 > 184	9	154 > 118	27
11	Aramit-1	17.238	175 > 135	15	175 > 107	25	N.D.	
	Aramit-2	17.604	175 > 135	15	175 > 107	25	N.D.	
12	Aspon	13.618	211 > 115	15	211 > 97	27	253 > 115	21
13	Atrazine	9.915	215 > 58	15	215 > 173	9	200 > 122	12
14	Azaconazole	17.301	217 > 173	21	173 > 145	18	217 > 145	30
15	Azinphos-ethyl	23.858	160 > 132	6	132 > 77	15	160 > 77	24
16	Azinphos-methyl	22.618	160 > 132	5	160 > 77	20	132 > 77	15
17	Benalaxyl	19.319	148 > 105	18	148 > 77	30	148 > 79	24
18	Benodanil	18.589	231 > 203	15	323 > 231	15	231 > 76	36
19	Benoxacor	11.467	120 > 93	20	120 > 77	20	120 > 65	20
20	Benzoylprop_ethyl	21.015	105 > 77	15	105 > 51	27	292 > 105	9
21	BHC_alpha	9.353	181 > 145	15	219 > 183	9	181 > 109	30
	BHC_beta	10.06	181 > 145	15	219 > 183	12	181 > 109	30
	BHC_delta	10.921	181 > 145	15	219 > 183	12	181 > 109	30
	BHC_gamma	10.252	181 > 145	15	219 > 183	12	181 > 109	30
22	Bifenox	22.007	341 > 310	10	341 > 281	12	N.D.	
23	Bifenthrin	21.574	181 > 166	15	181 > 165	30	181 > 179	12
24	Bromacil	13.053	207 > 190	15	205 > 188	15	205 > 162	15
25	Bromobutide	11.958	119 > 91	12	119 > 65	27	103 > 77	15
26	Bromophos-ethyl	15.768	359 > 303	15	303 > 285	15	242 > 97	30
27	Bromophos-methyl	14.304	331 > 316	15	125 > 79	6	331 > 286	27
28	Bromopropylate	21.378	183 > 155	15	341 > 183	15	341 > 185	20
29	Bupirimate	17.397	273 > 193	8	273 > 150	8	N.D.	
30	Butachlor	16.186	176 > 147	15	176 > 134	15	176 > 158	15
31	Butafenacil	25.678	331 > 180	18	180 > 124	21	180 > 152	10
32	Butralin	14.295	266 > 220	12	266 > 174	27	266 > 190	18
33	Butylate	6.153	156 > 57	12	146 > 90	9	146 > 57	12
34	Cadusafos	9.077	159 > 131	9	159 > 97	21	125 > 97	9
35	Captan	15.07	149 > 70	20	151 > 80	5	151 > 79	15
36	Carbophenothion	19.226	121 > 65	15	157 > 121	27	157 > 45	18
37	Chinomethionat	15.532	206 > 148	15	234 > 206	9	234 > 148	27
38	Chlorbenside	15.42	125 > 99	20	125 > 89	20	125 > 63	40
39	Chlorbufam	9.83	153 > 125	15	153 > 90	24	N.D.	
40	Chlordane_1	16.117	373 > 264	18	373 > 266	30	237 > 165	36
	Chlordane_2	15.604	373 > 264	18	373 > 266	30	237 > 165	36
41	Chlorethoxyfos	8.252	153 > 97	12	125 > 97	9	153 > 125	6
42	Chlorfenapyr	17.835	137 > 102	15	137 > 75	30	247 > 177	15
43	Chlorfenson	16.363	175 > 111	12	302 > 175	9	302 > 111	24
44	Chlorfluazuron	16.377	321 > 304	21	323 > 306	33	N.D.	
45	Chlorflurenol_methyl	15.533	215 > 152	27	152 > 151	24	215 > 187	15
46	Chlornitrofen	19.112	317 > 287	10	317 > 236	10	317 > 196	30
47	Chlorobenzilate	18.014	251 > 139	10	251 > 111	40	139 > 111	10
48	Chloroneb	6.935	206 > 191	12	206 > 141	21	193 > 113	18
49	Chloropropylate	18.014	139 > 111	12	251 > 111	26	139 > 75	26
50	Chloroxuron	12.566	245 > 182	10	245 > 154	25	245 > 111	25
51	Chlorpropham	8.546	213 > 171	10	213 > 127	5	127 > 100	15
52	Chlorpyrifos	13.738	197 > 169	15	199 > 171	15	314 > 258	18
53	Chlorpyrifos-methyl	12.158	286 > 93	24	125 > 79	6	286 > 271	15
54	Chlorthal-dimethyl	13.895	301 > 223	21	332 > 301	15	301 > 273	14
55	Chlorthalonil	11.05	266 > 231	20	266 > 168	30	266 > 133	40
56	Chlorthion	14.062	125 > 79	9	109 > 79	12	297 > 109	15
57	Chlorthiophos_1	18.647	325 > 269	15	325 > 324	6	269 > 177	21
	Chlorthiophos_2	18.647	325 > 269	15	269 > 205	18	325 > 205	30
58	Chlozolinate	14.974	187 > 124	21	331 > 259	6	187 > 159	12
59	Cinidone_ethyl	31.596	358 > 330	10	330 > 302	25	330 > 222	35
60	Cinmethylin	12.538	105 > 77	18	123 > 81	9	105 > 79	12
61	Clomazon	10.035	204 > 107	25	125 > 99	25	125 > 89	25
62	Clomeporp	22.119	148 > 120	5	288 > 120	25	288 > 169	10
63	Coumaphos	25.193	362 > 109	20	226 > 163	10	210 > 182	5
64	Crotoxyphos	15.422	193 > 127	20	166 > 127	5	127 > 109	20
65	Cyanazine	13.717	225 > 189	15	225 > 172	15	172 > 94	15
66	Cyanophos	10.365	109 > 79	9	125 > 79	9	243 > 109	9
67	Cycloate	8.4	83 > 55	9	154 > 83	9	154 > 55	24
68	Cyflufenamid	17.768	91 > 65	15	118 > 90	12	118 > 89	27
69	Cyfluthrin-1	25.995	163 > 127	9	163 > 91	21	226 > 199	9
	Cyfluthrin-2	26.178	163 > 127	9	163 > 91	21	226 > 199	9
	Cyfluthrin-3	26.324	163 > 127	9	163 > 91	21	226 > 199	9
	Cyfluthrin-4	26.404	163 > 127	9	163 > 91	21	226 > 199	9
70	Cyhalofop-butyl	23.058	256 > 120	10	229 > 109	15	120 > 91	15
71	Cyhalothrin-1	23.129	208 > 181	8	197 > 141	6	197 > 161	6
	Cyhalothrin-2	23.504	208 > 181	8	197 > 141	6	197 > 161	6
72	Cypermethrin-1	26.58	163 > 127	6	181 > 152	27	181 > 127	27
	Cypermethrin-2	26.773	163 > 127	6	181 > 152	27	181 > 127	27
	Cypermethrin-3	26.914	163 > 127	6	181 > 152	27	181 > 127	27
	Cypermethrin-4	26.987	163 > 127	6	181 > 152	27	181 > 127	27
73	Cyprazine	11.836	227 > 212	10	212 > 170	10	212 > 109	25
74	Cyproconazole-1	17.615	139 > 111	18	139 > 75	27	N.D.	
	Cyproconazole-2	17.615	139 > 111	18	222 > 125	27	139 > 75	27
75	Cyprodinil	14.554	224 > 208	18	224 > 118	36	210 > 93	15
76	DDD_pp	18.312	235 > 165	21	235 > 199	21	235 > 163	36
	DDE_pp	16.88	246 > 176	30	318 > 248	18	318 > 246	33
	DDT_op	18.422	235 > 165	27	235 > 199	15	237 > 165	24
	DDT_pp	19.599	235 > 165	18	235 > 199	12	237 > 165	27
77	Deltamethrin-1	29.329	181 > 152	24	253 > 93	24	253 > 172	9
	Deltamethrin-2	29.719	181 > 152	24	253 > 93	24	253 > 172	9
78	Demeton_O	8.157	171 > 115	10	115 > 97	10	88 > 60	5
79	Demeton_S	9.605	170 > 114	5	88 > 60	5	N.D.	
80	Demeton_S_methylsulfone	12.742	169 > 125	5	169 > 109	10	N.D.	
81	Desmetryn	11.73	213 > 58	12	213 > 171	6	198 > 108	12
82	Diafor	24.099	208 > 89	25	173 > 104	10	N.D.	
83	Diallate-1	9.177	234 > 150	21	128 > 86	6	234 > 192	15
	Diallate-2	9.41	234 > 150	21	234 > 192	15	128 > 86	6
84	Diazinon	10.708	137 > 84	15	137 > 54	27	179 > 137	18
85	Dichlofenthion	11.865	279 > 223	15	223 > 205	15	223 > 159	21
86	Dichlofluanid	13.294	123 > 77	18	224 > 123	20	224 > 77	40
87	Dichlormid	5.591	172 > 108	9	172 > 56	18	172 > 96	9
88	Dichlorvos	4.796	109 > 79	9	185 > 93	15	185 > 63	24
89	Diclobutrazole	17.235	272 > 161	10	270 > 201	5	270 > 159	10
90	Diclofop_methyl	20.224	340 > 253	15	253 > 162	20	340 > 281	15
91	Dicloran	9.644	206 > 176	15	176 > 148	12	206 > 124	27
92	Dicofol	13.792	139 > 111	18	139 > 75	30	250 > 139	15
93	Dieldrin	16.896	263 > 193	27	279 > 243	15	263 > 203	21
94	Diethatyl-ethyl	16.463	188 > 160	9	188 > 131	21	188 > 130	33
95	Diethofencarb	13.532	124 > 96	9	151 > 123	12	151 > 77	24
96	Difenoconazole-1	29.057	323 > 265	15	323 > 202	35	265 > 202	20
	Difenoconazole-2	29.183	323 > 265	15	323 > 202	35	265 > 202	20
97	Diflufenican	20.35	394 > 266	18	266 > 246	15	266 > 183	21
98	Dimepiperate	15.181	119 > 91	12	103 > 77	12	145 > 69	18
99	Dimethachlor	11.891	134 > 105	15	134 > 77	30	197 > 148	9
100	Dimethametryn	14.8	212 > 122	15	212 > 142	15	255 > 212	20
101	Dimethenamid	11.93	230 > 154	12	154 > 111	12	203 > 126	21
102	Dimethoate	9.669	143 > 111	10	125 > 79	5	125 > 47	20
103	Dimethylvinphos-(E)	13.231	295 > 109	12	297 > 109	21	N.D.	
	Dimethylvinphos-(Z)	13.682	295 > 109	12	297 > 109	21	N.D.	
104	Diniconazole	18.194	268 > 232	15	268 > 136	33	232 > 150	21
105	Dinitramine	10.922	261 > 195	24	261 > 241	6	305 > 230	12
106	Dioxathion	10.25	125 > 97	9	97 > 65	18	97 > 79	15
107	Diphenamid	14.334	167 > 152	18	167 > 165	30	152 > 151	21
108	Diphenylamine	8.24	169 > 66	21	169 > 77	30	169 > 141	27
109	Dithiopyr	12.948	354 > 286	15	354 > 306	12	306 > 286	9
110	Edifenphos	19.364	173 > 109	12	109 > 65	18	109 > 69	27
111	Endosulfan_alphs	16.012	195 > 159	12	195 > 160	15	195 > 125	27
112	Endosulfan_beta	17.957	195 > 159	6	195 > 160	9	195 > 125	27
	Endosulfan_sulfate	19.436	272 > 165	36	237 > 116	18	N.D.	
113	Endrin	17.633	263 > 191	27	263 > 193	30	263 > 228	21
114	EPN	21.405	169 > 141	9	169 > 77	27	157 > 110	15
115	Epoxiconazole	20.688	192 > 138	12	192 > 111	21	192 > 102	30
116	EPTC	5.561	128 > 86	6	189 > 128	6	189 > 86	12
117	Esprocarb	13.157	222 > 91	15	91 > 65	18	222 > 151	6
118	Etaconazole_1	18.221	173 > 145	18	173 > 109	27	245 > 55	15
	Etaconazole_2	18.35	173 > 145	15	173 > 109	27	245 > 55	15
119	Ethalfluralin	8.657	276 > 202	18	276 > 105	30	316 > 276	6
120	Ethion	18.543	231 > 129	24	153 > 97	12	231 > 175	15
121	Ethofumesate	13.133	161 > 105	12	161 > 133	6	286 > 207	9
122	Ethoprophos	8.372	200 > 158	6	158 > 97	15	158 > 114	9
123	Ethychlozate	14.838	165 > 102	20	165 > 111	25	165 > 138	20
124	Etofenprox	27.124	163 > 135	12	163 > 107	21	135 > 107	10
125	Etoxazole	21.853	141 > 113	15	141 > 63	24	300 > 270	21
126	Etridiazole	6.442	211 > 140	24	211 > 183	12	N.D.	
127	Etrimfos	11.163	181 > 153	12	181 > 56	24	292 > 181	15
128	Fenamidone	21.879	238 > 103	27	268 > 180	24	238 > 91	28
129	Fenamiphos	16.42	303 > 288	10	303 > 260	15	303 > 195	10
130	Fenarimol	23.564	139 > 111	15	139 > 75	30	107 > 79	9
131	Fenazaquin	21.934	145 > 117	12	160 > 145	9	145 > 91	30
132	Fenbuconazole	25.872	198 > 129	9	129 > 102	15	129 > 78	18
133	Fenchlorphos	12.616	285 > 270	15	125 > 79	6	285 > 93	27
134	Fenclorim	9.332	224 > 189	15	224 > 104	35	189 > 104	20
135	Fenfuram	10.89	201 > 109	25	109 > 53	20	N.D.	
136	Fenitrothion	13.034	125 > 79	9	277 > 109	21	260 > 125	12
137	Fenobucarb	8.042	121 > 77	21	150 > 121	9	121 > 103	15
138	Fenothiocarb	15.856	161 > 72	10	161 > 55	16	N.D.	
139	Fenoxanil	17.851	189 > 125	15	189 > 154	12	189 > 109	33
140	Fenoxycarb	21.409	255 > 186	10	186 > 158	5	186 > 109	12
141	Fenpropathrin	21.769	97 > 55	9	181 > 152	24	265 > 210	12
142	Fenson	14.086	141 > 77	9	141 > 51	30	99 > 73	12
143	Fenthion	13.666	278 > 109	21	278 > 169	15	278 > 125	21
144	Fenvalerate-1	28.316	167 > 125	15	225 > 119	18	225 > 147	9
	Fenvalerate-2	28.709	167 > 125	15	225 > 119	18	225 > 147	9
145	Fipronil	15.17	367 > 213	27	351 > 255	18	367 > 254	21
146	Flamprop-isopropyl	18.329	105 > 77	15	105 > 51	27	276 > 105	12
147	Flonicamid	7.892	174 > 146	15	174 > 126	25	N.D.	
148	Fluazifop_butyl	17.819	282 > 238	20	282 > 91	20	N.D.	
149	Fluchloralin	10.797	306 > 264	9	326 > 63	12	306 > 160	27
150	Flucythrinate_1	27.017	199 > 157	10	199 > 107	35	157 > 107	20
	Flucythrinate_2	27.395	199 > 157	10	199 > 107	35	157 > 107	20
151	Fludioxonil	16.81	248 > 182	15	248 > 154	15	248 > 127	35
152	Flufenpyr_ethyl	18.185	408 > 345	15	373 > 345	10	321 > 286	15
153	Flumetralin	16.212	143 > 107	27	143 > 108	21	157 > 129	15
154	Flumiclorac_pentyl	30.025	318 > 260	15	318 > 107	35	308 > 280	10
155	Flumioxazine	28.346	354 > 326	10	354 > 176	15	287 > 259	15
156	Fluopyram	15.15	173 > 145	18	173 > 95	30	145 > 95	12
157	Fluorodifen	16.536	190 > 146	6	190 > 126	9	190 > 75	21
158	Flurochloridone	14.113	311 > 174	20	311 > 103	20	187 > 159	15
159	Flusilazole	17.272	233 > 165	18	233 > 152	18	233 > 91	21
160	Flutamone	22.537	333 > 120	15	199 > 157	20	N.D.	
161	Fluthiacet_methyl	31.849	405 > 56	15	403 > 84	10	403 > 56	15
162	Flutianil	27.765	231 > 216	5	231 > 200	15	200 > 199	10
163	Flutolanil	16.576	173 > 145	15	281 > 173	9	173 > 95	30
164	Flutriafol	16.248	123 > 95	15	123 > 75	27	219 > 123	15
165	Fluvalinate-1	28.736	250 > 55	15	250 > 200	15	252 > 55	20
	Fluvalinate-2	28.861	250 > 55	15	250 > 200	15	252 > 55	20
166	Folpet	15.293	104 > 76	15	260 > 130	18	262 > 130	18
167	Fonofos	10.481	137 > 109	9	109 > 81	9	109 > 65	12
168	Formothion	11.474	224 > 196	10	224 > 125	15	125 > 79	15
169	Fosthiazate-1	14.287	195 > 103	9	195 > 60	30	139 > 75	14
	Fosthiazate-2	14.374	195 > 103	9	195 > 60	30	139 > 75	14
170	Fthalide	14.202	272 > 243	20	272 > 215	40	243 > 215	20
171	Furathiocarb	22.528	163 > 107	15	163 > 135	6	194 > 161	12
172	Halfenprox	26.656	263 > 115	24	263 > 117	12	263 > 129	39
173	Heptachlor	12.394	100 > 65	15	272 > 237	12	274 > 239	16
174	Heptachlor-epoxide	14.807	353 > 263	14	353 > 282	12	353 > 317	10
175	Heptenophos	7.686	124 > 89	15	124 > 63	30	89 > 63	21
176	Hexachlorbenzene	9.57	284 > 249	24	284 > 214	27	249 > 214	15
177	Hexaconazole	16.503	83 > 82	9	214 > 159	18	214 > 152	30
178	Imazalil	16.643	215 > 173	9	173 > 109	30	N.D.	
179	Indanofan	21.915	174 > 159	9	159 > 103	15	159 > 77	30
180	Indoxacarb	29.706	203 > 134	12	203 > 106	21	264 > 176	14
181	Iprobenfos	11.345	204 > 91	9	91 > 65	21	204 > 121	33
182	Iprodione	21.068	187 > 124	24	314 > 56	24	314 > 245	15
183	Iprovalicarb-1	17.096	134 > 42	20	119 > 91	15	116 > 98	5
	Iprovalicarb-2	17.445	134 > 42	20	119 > 91	15	116 > 98	5
184	Isazofos	11.107	161 > 119	9	119 > 76	24	161 > 146	9
185	Isofenphos	15.139	213 > 121	18	213 > 185	6	213 > 65	33
186	Isofenphos-methyl	14.62	199 > 121	15	121 > 65	18	199 > 93	27
187	Isopropalin	14.56	280 > 238	9	280 > 165	21	280 > 180	12
188	Isoprothiolane	16.705	162 > 134	9	162 > 85	21	118 > 90	15
189	Isotianil	19.626	180 > 91	15	297 > 180	20	297 > 262	5
190	Isoxadifen-ethyl	19.112	222 > 204	20	222 > 178	20	204 > 176	20
191	Isoxanthion	17.581	177 > 130	12	177 > 116	12	N.D.	
192	Kresoxim-methyl	17.453	116 > 89	15	206 > 131	15	206 > 116	6
193	Lactofen	23.617	223 > 132	24	344 > 223	27	344 > 179	30
194	Leptophos	22.716	377 > 362	24	377 > 269	36	375 > 360	24
195	Malathion	13.398	127 > 99	9	173 > 99	15	173 > 127	6
196	Mecarbam	15.188	97 > 65	18	131 > 74	18	131 > 86	15
197	Mefenacet	23.003	192 > 136	20	192 > 109	30	N.D.	
198	Mefenpyr-diethyl	20.909	253 > 189	21	253 > 190	12	299 > 253	24
199	Mepronil	18.782	119 > 91	15	119 > 65	35	N.D.	
200	Metazachlor	14.749	209 > 133	10	277 > 133	10	277 > 209	10
201	Metconazole	21.914	125 > 89	27	125 > 63	27	138 > 69	12
202	Methidathion	15.66	145 > 85	18	145 > 58	21	125 > 79	9
203	Methoprotryne	17.303	256 > 212	15	256 > 170	27	256 > 158	21
204	Methoxychlor	21.644	227 > 169	30	227 > 141	36	227 > 212	14
205	Methyltrithion	17.888	125 > 79	15	157 > 121	27	157 > 75	36
206	Metolachlor	13.582	162 > 133	15	238 > 162	15	162 > 132	24
207	Metrafenone	24.211	393 > 362	24	377 > 347	30	377 > 362	12
208	Metribuzin	11.909	198 > 82	18	198 > 89	15	198 > 110	10
209	MGK-264_1	14.251	164 > 93	15	164 > 98	18	164 > 80	30
	MGK-264_2	14.632	164 > 98	15	164 > 67	9	164 > 80	27
210	Mirex	22.904	272 > 237	15	272 > 143	40	272 > 119	40
211	Molinate	7.283	126 > 55	15	187 > 126	6	N.D.	
212	Monolinuron	9.947	214 > 61	10	126 > 99	10	N.D.	
213	Myclobutanil	17.147	179 > 125	15	179 > 90	30	150 > 123	18
214	Napropamide	16.451	128 > 72	6	100 > 72	6	128 > 100	12
215	Nitrapyrin	6.44	194 > 133	18	194 > 112	30	194 > 158	24
216	Nitrothal-isopropyl	14.015	236 > 194	9	236 > 148	18	254 > 212	10
217	Nonachlor_cis	18.419	409 > 300	30	409 > 109	24	409 > 302	27
218	Nonachlor_trans	16.287	409 > 300	21	409 > 263	30	407 > 300	24
219	Norflurazon	19.496	303 > 145	20	303 > 173	15	303 > 102	30
220	Nuarimol	20.076	139 > 111	15	235 > 139	18	139 > 75	27
221	Ofurace	19.132	232 > 158	18	132 > 117	15	232 > 186	12
222	Oxadiazon	17.077	258 > 175	10	258 > 147	15	258 > 112	35
223	Oxadixyl	18.471	163 > 132	9	132 > 117	18	163 > 117	27
224	Oxyflofen	17.273	361 > 317	5	361 > 300	5	N.D.	
225	Paclobutrazole	15.823	236 > 125	12	125 > 89	27	236 > 103	24
226	Parathion-ethyl	13.76	291 > 109	21	139 > 109	6	291 > 81	24
227	Parathion-methyl	12.149	263 > 109	18	263 > 137	15	263 > 246	6
228	Pebulate	6.428	128 > 57	9	128 > 72	6	161 > 128	6
229	Penconazole	14.84	159 > 123	18	248 > 157	24	159 > 89	30
230	Pendimethalin	14.812	252 > 162	12	252 > 161	21	162 > 147	9
231	Penflufen	18.919	274 > 141	15	141 > 84	20	141 > 60	20
232	Pentachlorobezonitrile	10.449	275 > 240	10	275 > 205	35	N.D.	
233	Penthiopyrad	18.517	177 > 101	20	177 > 149	25	177 > 75	25
234	Pentoxazon	22.677	285 > 70	15	287 > 70	10	187 > 131	10
235	Permethrin-1	24.821	183 > 153	18	183 > 168	12	183 > 165	14
	Permethrin-2	25.071	183 > 153	18	183 > 168	12	183 > 165	14
236	Perthane	17.768	223 > 193	40	223 > 179	30	223 > 165	30
237	Phenothrin	22.481	123 > 81	5	183 > 115	40	183 > 168	15
238	Phenthoate	15.234	274 > 121	15	274 > 125	18	274 > 246	9
239	Phorate	9.194	260 > 75	10	231 > 129	25	121 > 65	10
240	Phosalone	22.638	182 > 111	18	182 > 75	30	121 > 65	12
241	Phosmet(PMP)	21.257	160 > 77	24	160 > 133	15	160 > 105	18
242	Phosphamidone	11.869	127 > 109	0	127 > 95	0	264 > 127	0
243	Picolinafen	21.521	376 > 238	30	238 > 145	18	376 > 239	6
244	Picoxystrobin	16.495	145 > 102	24	145 > 115	15	335 > 173	12
245	Piperonyl butoxide	20.486	176 > 131	15	176 > 145	15	176 > 117	25
246	Piperophos	21.614	320 > 122	12	140 > 98	15	140 > 81	18
247	Pirimiphos-ethyl	14.502	318 > 166	18	318 > 182	15	333 > 168	21
248	Pirimiphos-methyl	13.11	290 > 125	21	290 > 233	12	290 > 151	21
249	Pretilachlor	16.887	238 > 162	15	238 > 146	15	162 > 147	10
250	Primicarb	11.492	238 > 166	10	166 > 96	10	166 > 71	30
251	Probenazole	10.227	159 > 130	10	130 > 103	18	130 > 77	30
252	Prochloraz	25.306	180 > 138	12	180 > 69	18	308 > 70	18
253	Procymidone	15.406	96 > 67	9	96 > 53	18	283 > 96	9
254	Profenofos	16.761	339 > 269	18	208 > 99	21	339 > 188	30
255	Profluralin	10.415	318 > 199	15	318 > 55	18	330 > 69	21
256	Prometon	9.776	210 > 168	9	168 > 126	9	210 > 112	15
257	Prometryn	12.526	241 > 226	15	241 > 199	10	241 > 184	15
258	Pronamide	10.435	173 > 145	15	173 > 109	27	145 > 109	15
259	Propachlor	8.133	120 > 77	21	176 > 57	9	120 > 51	27
260	Propanil	11.803	217 > 161	15	161 > 126	15	161 > 99	35
261	Propazine	10.039	214 > 172	12	172 > 69	21	172 > 94	15
262	Propetamphos	10.353	138 > 110	9	138 > 64	18	194 > 166	9
263	Propham	6.374	119 > 91	12	119 > 64	24	91 > 64	12
264	Propiconazole-1	19.485	259 > 191	10	259 > 173	20	259 > 69	30
	Propiconazole-2	19.706	259 > 191	10	259 > 173	20	259 > 69	30
265	Propisochlor	12.533	162 > 120	15	132 > 117	12	162 > 91	30
266	Prothiophos	16.66	267 > 239	9	309 > 239	15	267 > 205	30
267	Pyracabolid	14.215	125 > 107	5	125 > 97	10	125 > 55	15
268	Pyraclofos	24.093	194 > 138	21	194 > 139	15	360 > 194	12
269	Pyrazophos	23.888	221 > 193	9	232 > 204	9	221 > 177	21
270	Pyridaben	25.015	147 > 117	21	147 > 132	15	147 > 119	10
271	Pyridalyl	27.252	204 > 176	10	204 > 148	20	204 > 146	30
272	Pyridaphenthion	21.221	340 > 199	15	340 > 109	20	199 > 92	10
273	Pyrifenox 1	14.947	171 > 100	24	171 > 136	12	262 > 91	24
	Pyrifenox 2	15.828	171 > 100	27	171 > 136	12	262 > 91	21
274	Pyrimidifen	27.977	184 > 169	15	184 > 157	15	186 > 171	20
275	Pyriminobac-methyl(E)	19.936	302 > 256	21	302 > 230	18	330 > 254	10
276	Quinalphos	15.209	146 > 118	9	146 > 91	27	157 > 129	15
277	Quinoxyfen	19.373	237 > 208	27	272 > 237	18	237 > 181	39
278	Quintozene	10.393	295 > 237	16	265 > 237	10	295 > 265	12
279	Sectumeton	10.886	169 > 154	9	196 > 85	9	196 > 57	27
280	Silafluofen	27.439	286 > 258	10	286 > 179	10	286 > 165	25
281	Simeconazole	12.259	121 > 101	15	121 > 75	24	195 > 75	18
282	Simetryn	12.244	213 > 198	5	213 > 185	10	213 > 170	10
283	Spiromesifen	21.038	272 > 209	20	272 > 254	15	N.D.	
284	Spiroxamine 2	12.103	100 > 72	9	100 > 58	12	100 > 99	14
284	Spiroxamine_1	12.972	100 > 72	9	100 > 58	12	100 > 99	14
285	Sulfotep	9.042	238 > 146	15	322 > 146	27	322 > 202	9
286	Sulprofos	18.918	156 > 141	18	322 > 156	12	322 > 139	15
287	TCMTB	16.255	180 > 136	15	238 > 180	5	180 > 109	30
288	Tebuconazole	20.057	125 > 89	18	250 > 125	24	125 > 90	24
289	Tebufenpyrad	21.873	333 > 171	18	171 > 88	21	333 > 276	9
290	Tebupirimfos	11.317	261 > 137	15	234 > 110	12	234 > 126	12
291	Tefluthrin	11.026	177 > 127	18	197 > 141	12	177 > 137	16
292	Terbacil	10.869	161 > 144	15	161 > 88	24	N.D.	
293	Terbufos	10.375	231 > 129	27	231 > 175	12	231 > 203	9
294	Terbumeton	10.081	169 > 154	9	169 > 112	15	169 > 69	30
295	Terbuthylazine	10.336	214 > 71	15	214 > 104	21	214 > 132	15
296	Terbutryn	12.938	241 > 170	15	226 > 136	15	226 > 96	15
297	Tetrachlorvinphos	16.033	329 > 109	21	109 > 79	9	331 > 109	21
298	Tetraconazole	14.023	336 > 156	27	336 > 204	33	336 > 183	24
299	Tetradifon	22.284	111 > 75	15	159 > 131	12	159 > 111	21
300	Tetramethrin-1	21.301	164 > 107	15	164 > 77	27	123 > 81	9
	Tetramethrin-2	21.548	164 > 107	12	164 > 77	27	123 > 81	9
301	Tetrasul	18.685	324 > 254	18	324 > 252	33	252 > 182	30
302	Thiazopyr	13.689	327 > 277	27	327 > 292	21	327 > 252	36
303	Thifluzamide	17.427	166 > 125	15	449 > 429	21	447 > 427	21
304	Thiometon	9.48	125 > 47	14	125 > 79	10	125 > 63	8
305	Thionazin	8.033	143 > 79	10	143 > 52	35	N.D.	
306	Tolclofos_methyl	12.305	265 > 250	15	265 > 93	27	265 > 220	21
307	Tolfenpyrad	30.404	383 > 171	30	383 > 181	5	385 > 173	35
308	Tolylfluanid	14.99	137 > 91	18	238 > 137	12	137 > 65	30
309	Tralomethrin-1	29.326	181 > 152	27	253 > 93	21	253 > 174	9
	Tralomethrin-2	29.718	181 > 152	27	253 > 93	21	253 > 174	9
310	Triadimefon	13.834	208 > 181	12	208 > 127	15	208 > 111	24
311	Triadimenol	15.209	168 > 70	12	112 > 58	9	128 > 65	22
312	Triallate	11.119	268 > 226	15	268 > 184	30	143 > 83	15
313	Triazophos	18.978	161 > 134	9	161 > 106	15	257 > 162	9
314	Tribufos	16.89	169 > 57	9	202 > 147	9	202 > 113	21
315	Tridiphane	12.588	187 > 159	20	187 > 123	35	173 > 145	20
316	Triflumizole	15.541	206 > 179	18	278 > 73	6	206 > 144	24
317	Triflumuron	6.4	139 > 111	18	139 > 75	27	N.D.	
318	Trifluralin	8.866	306 > 264	9	264 > 160	18	264 > 206	9
319	Uniconazole	16.852	234 > 165	9	234 > 102	30	234 > 137	24
320	Vernlolate	6.296	128 > 86	6	161 > 160	12	161 > 128	9
321	Vinclozoline	12.141	198 > 145	18	212 > 172	15	198 > 109	27
322	Zoxamide	15.344	187 > 159	15	187 > 123	27	242 > 186	21

RT: Retention Time; N.D.: Not Detected; CE: Collision Energy.

**Table 3 foods-12-03001-t003:** Collected samples.

Agricultural Product	Number of Samples
Chili pepper	21
Carrot	17
Stem of garlic	7
Mango	10
Wheat	14
Banana	24
Almond	10
Cabbage	10
Coffee bean	12
Pineapple	10

**Table 4 foods-12-03001-t004:** Average recovery (Ave., %) and SD (%) for the GC-MS/MS method applied to the studied samples (*n* = 5) at spiked level (0.01 mg/kg).

	Coffee		Potato		Maize		Chili Pepper	
Compound	Ave. (%)	SD	Ave. (%)	SD	Ave. (%)	SD	Ave. (%)	SD
2,6-Diisoporpylnaphthalene	69.1	8.6	72.4	2.4	65.3	2.0	74.5	1.7
Acetochlor	75.9	5.1	74.3	4.8	76.6	3.0	79.9	1.6
Acibenzola_s_methyl	92.4	12.3	66.4	1.6	73.8	1.7	83.7	11.9
Acrinathrin_1	86.5	3.8	77.6	4.0	73.0	5.1	95.7	4.1
Acrinathrin_2	81.5	3.7	77.2	8.1	76.1	7.8	102.3	8.1
Alachlor	79.2	5.8	73.7	2.1	76.0	3.0	84.9	0.5
Aldrin	65.2	7.6	63.8	1.0	54.9	1.8	76.7	4.0
Allethrin-1	60.5	16.6	86.1	5.9	78.0	1.1	65.6	10.8
Allethrin-2	83.8	10.0	77.3	5.4	72.8	4.1	75.3	9.5
Allidochlor	72.3	4.2	61.2	5.0	61.4	1.4	70.1	6.6
Ametryn	78.8	9.7	72.4	3.7	70.9	2.3	84.8	4.8
Anilofos	76.2	5.9	70.7	3.2	74.2	2.7	103.6	10.1
Aramit-1	92.7	3.0	89.1	9.3	66.5	3.4	84.2	5.2
Aramit-2	66.4	1.4	73.9	2.4	77.1	0.9	102.8	7.4
Aspon	77.7	8.4	73.1	2.6	73.2	2.5	87.7	6.9
Atrazine	73.5	8.7	69.7	2.3	70.6	1.6	70.9	7.0
Azaconazole	76.5	9.2	70.7	2.1	68.3	3.4	79.3	2.9
Azinphos-ethyl	75.6	4.7	72.6	4.8	74.5	6.3	104.5	9.3
Azinphos-methyl	93.5	3.4	67.8	6.4	70.5	3.9	97.6	11.8
Benalaxyl	82.6	7.8	77.4	1.7	74.5	3.0	84.8	4.7
Benodanil	84.7	9.6	72.7	3.4	71.2	5.0	98.9	4.9
Benoxacor	125.6	56.0	70.7	2.3	73.9	2.2	84.5	3.4
Benzoylprop_ethyl	82.9	6.5	76.9	2.2	75.0	1.7	85.1	4.7
BHC_alpha	69.5	5.6	67.7	2.5	66.6	2.5	79.3	4.6
BHC_beta	71.6	6.0	70.1	1.3	68.4	2.1	81.1	2.0
BHC_delta	69.8	6.2	70.2	2.7	68.8	2.7	80.6	5.5
BHC_gamma	71.9	4.9	67.9	2.0	66.6	2.5	75.7	1.6
Bifenox	86.6	3.6	61.5	3.6	70.1	4.6	104.0	8.7
Bifenthrin	77.5	8.6	76.5	2.5	68.4	1.7	88.0	4.6
Bromacil	74.4	11.3	75.1	1.7	77.1	2.5	82.4	1.1
Bromobutide	81.5	6.9	72.8	2.9	73.5	3.7	81.1	4.4
Bromophos-ethyl	74.7	9.4	70.6	3.0	68.5	2.2	83.8	8.5
Bromophos-methyl	76.4	7.4	71.6	4.5	71.2	2.8	84.6	1.2
Bromopropylate	80.1	7.4	76.7	4.0	72.6	1.7	89.9	6.7
Bupirimate	81.7	5.6	71.6	3.6	75.9	3.8	90.1	5.2
Butachlor	83.9	10.6	77.0	1.3	74.2	5.5	90.2	1.9
Butafenacil	82.5	4.8	76.9	5.4	70.7	2.2	99.6	8.0
Butralin	84.2	7.4	71.4	2.4	74.5	4.4	79.2	2.6
Butylate	69.0	5.3	58.6	1.4	60.7	1.7	61.7	2.0
Cadusafos	77.7	4.0	72.0	3.6	73.2	2.7	82.8	3.0
Captan	79.8	3.2	78.1	7.4	72.9	1.4	55.1	39.0
Carbophenothion	78.5	11.6	71.6	3.0	67.3	3.5	91.7	4.2
Chinomethionat	71.3	7.6	68.7	2.5	65.5	3.2	64.6	6.6
Chlorbenside	70.2	11.4	64.5	5.2	59.6	4.4	97.2	7.3
Chlorbufam	69.0	8.3	71.9	4.3	74.4	0.8	86.7	2.2
Chlordane_1	73.5	11.1	71.7	1.5	62.3	1.1	79.3	5.3
Chlordane_2	70.1	11.4	68.0	1.2	65.8	3.0	79.2	11.9
Chlorethoxyfos	72.7	3.9	67.9	2.6	68.4	2.1	79.2	2.2
Chlorfenapyr	75.8	4.2	73.0	4.6	72.4	5.0	86.6	1.0
Chlorfenson	77.1	9.1	72.9	3.0	70.4	2.3	75.9	10.7
Chlorfluazuron	26.4	12.1	76.2	11.9	70.0	8.7	63.5	7.9
Chlorflurenol_methyl	80.4	7.3	70.6	2.6	72.4	2.7	81.8	1.4
Chlornitrofen	87.5	5.7	69.1	4.1	67.6	7.3	86.8	4.7
Chlorobenzilate	82.9	7.0	72.5	2.9	69.6	4.1	91.9	6.2
Chloroneb	72.6	6.0	67.8	2.4	66.0	1.1	75.6	3.0
Chloropropylate	80.2	8.2	71.5	3.1	70.1	3.9	88.3	4.0
Chloroxuron	65.5	4.7	70.5	3.3	67.1	3.5	84.0	3.0
Chlorpropham	74.8	7.8	76.0	1.9	74.1	1.8	78.1	6.6
Chlorpyrifos	82.2	6.3	74.8	3.4	72.5	2.8	83.7	2.1
Chlorpyrifos-methyl	76.4	6.7	71.2	1.6	73.4	1.0	82.5	1.2
Chlorthal-dimethyl	83.6	6.4	76.3	3.4	75.4	1.5	81.9	4.0
Chlorthalonil	77.2	11.4	83.1	2.2	73.9	11.4	28.0	36.4
Chlorthion	83.1	5.8	66.8	4.3	71.5	4.2	86.0	6.3
Chlorthiophos_1	74.2	9.5	72.1	2.6	72.3	2.8	87.7	4.6
Chlorthiophos_2	74.7	9.2	73.4	2.6	71.6	2.1	86.6	3.8
Chlozolinate	80.5	6.8	72.1	2.6	71.0	3.9	79.6	7.4
Cinidone_ethyl	84.0	8.3	79.3	3.9	78.3	5.5	93.8	8.5
Cinmethylin	83.2	8.3	82.3	3.4	77.0	3.4	91.5	2.4
Clomazon	79.4	8.2	72.5	3.8	72.7	1.7	77.8	2.3
Clomeporp	82.6	3.4	71.6	1.6	73.8	2.8	92.3	5.6
Coumaphos	75.1	4.8	72.9	4.1	72.8	2.8	99.6	8.0
Crotoxyphos	80.2	8.5	72.5	5.6	72.8	4.1	78.7	4.3
Cyanazine	79.4	8.9	72.8	3.1	74.2	3.0	89.2	6.9
Cyanophos	79.3	4.6	72.6	3.3	73.8	2.6	91.3	2.9
Cycloate	72.4	5.2	67.5	2.2	68.2	2.5	76.7	2.7
Cyflufenamid	82.6	8.9	76.4	3.7	75.9	5.0	88.2	1.1
Cyfluthrin-1	76.2	7.3	78.6	4.5	69.1	1.3	89.1	4.5
Cyfluthrin-2	78.5	6.5	78.2	4.3	72.2	2.0	91.4	3.8
Cyfluthrin-3	77.0	7.7	78.2	2.7	72.9	2.9	95.8	10.0
Cyfluthrin-4	80.3	6.7	77.4	4.4	69.7	2.5	96.2	2.2
Cyhalofop-butyl	79.6	5.9	77.9	2.9	75.9	1.7	88.2	4.4
Cyhalothrin-1	82.0	4.7	78.4	3.5	71.6	2.9	89.7	6.1
Cyhalothrin-2	86.3	1.7	74.5	2.5	71.6	4.0	92.7	9.7
Cypermethrin-1	79.9	7.3	80.1	4.4	66.9	2.4	92.7	6.3
Cypermethrin-2	80.4	6.9	79.4	3.9	68.7	3.8	98.2	6.7
Cypermethrin-3	82.1	2.8	79.3	4.9	67.7	2.2	91.5	7.2
Cypermethrin-4	81.6	7.1	75.9	2.7	67.6	4.2	99.5	9.6
Cyprazine	79.8	6.6	72.3	2.2	71.3	4.9	91.7	8.0
Cyproconazole-1	79.3	8.2	72.4	2.9	70.5	4.7	88.0	7.4
Cyproconazole-2	79.2	7.2	71.8	1.9	69.6	5.7	84.8	6.3
Cyprodinil	79.6	8.4	73.3	2.5	73.1	3.2	88.5	4.5
DDD_pp	72.3	7.7	67.6	2.6	65.5	2.1	82.8	2.5
DDE_pp	62.3	8.8	69.3	2.8	59.8	2.3	87.5	5.9
DDT_op	65.2	8.9	68.5	1.9	59.5	2.5	74.4	4.5
DDT_pp	66.8	8.5	70.2	1.9	61.4	1.4	83.0	3.7
Deltamethrin-1	116.4	6.4	67.6	6.1	57.4	5.2	88.0	30.7
Deltamethrin-2	80.3	6.3	77.4	4.4	72.5	3.9	95.7	7.2
Demeton_O	76.3	4.2	53.8	8.0	61.3	2.0	64.7	7.8
Demeton_S	71.9	7.7	49.8	5.4	64.2	4.7	67.0	13.5
Demeton_S_methylsulfone	23.8	20.0	6.8	4.0	4.5	3.2	0.4	0.3
Desmetryn	76.5	5.9	69.3	2.5	72.5	3.9	82.2	0.4
Diafor	78.8	1.9	75.2	5.1	75.0	4.3	104.4	8.4
Diallate-1	74.0	8.2	68.5	2.7	71.6	2.4	81.4	4.1
Diallate-2	71.7	7.3	68.9	3.8	68.4	1.5	85.0	1.1
Diazinon	74.6	7.5	74.6	1.6	72.6	3.1	79.2	2.6
Dichlofenthion	75.3	5.6	71.6	3.9	70.6	1.7	84.5	4.7
Dichlofluanid	80.5	7.5	78.6	4.1	73.1	2.9	60.8	9.3
Dichlormid	74.4	7.4	61.5	3.7	65.0	0.8	63.6	3.4
Dichlorvos	62.7	9.4	62.1	7.4	52.8	1.0	49.2	2.1
Diclobutrazole	83.7	6.4	69.8	3.2	71.0	5.5	67.9	14.5
Diclofop_methyl	79.1	8.6	78.2	2.2	77.2	1.0	85.9	4.1
Dicloran	73.0	6.0	68.8	3.2	72.5	3.6	88.0	0.5
Dicofol	73.8	8.0	71.1	2.5	67.0	2.2	91.7	2.5
Dieldrin	73.6	5.7	68.2	6.1	65.5	3.6	73.4	1.9
Diethatyl-ethyl	83.5	8.9	73.8	2.6	72.8	7.0	86.0	6.6
Diethofencarb	79.4	9.3	75.1	2.5	72.0	3.2	91.2	4.2
Difenoconazole-1	80.7	5.6	72.8	3.4	72.6	2.0	115.7	20.1
Difenoconazole-2	77.7	4.8	71.9	2.5	72.9	1.3	113.4	20.1
Diflufenican	80.7	6.6	75.9	1.7	75.0	1.6	87.5	2.4
Dimepiperate	77.6	8.3	72.0	1.9	73.2	4.0	86.0	7.3
Dimethachlor	78.2	6.3	73.8	3.2	73.8	2.8	85.1	3.0
Dimethametryn	79.5	8.9	72.3	2.9	72.4	2.0	82.0	4.3
Dimethenamid	79.6	7.4	73.2	2.7	74.9	2.1	81.9	2.9
Dimethoate	78.1	7.6	69.2	4.3	92.8	37.9	92.0	13.8
Dimethylvinphos-(E)	79.4	6.0	70.0	3.8	72.5	2.8	89.8	1.5
Dimethylvinphos-(Z)	79.6	5.6	70.7	3.7	74.4	4.3	84.3	5.7
Diniconazole	78.6	4.7	70.2	2.9	71.3	5.6	96.5	13.5
Dinitramine	79.0	6.0	66.4	2.2	72.3	2.4	83.7	9.2
Dioxathion	80.4	6.2	73.6	2.9	75.3	2.5	93.2	5.5
Diphenamid	79.8	7.7	72.8	2.8	73.6	1.6	85.4	5.6
Diphenylamine	75.8	5.9	68.4	3.1	69.8	2.1	66.4	1.4
Dithiopyr	82.0	7.8	73.9	2.0	77.2	1.9	85.6	2.8
Edifenphos	73.7	5.6	74.6	3.6	74.4	3.7	87.3	3.1
Endosulfan_alphs	213.1	26.4	69.5	2.1	66.4	7.2	95.4	2.9
Endosulfan_beta	77.7	9.3	67.0	5.3	63.5	4.1	76.9	5.1
Endosulfan_sulfate	73.2	8.7	66.2	6.6	61.3	6.1	86.1	12.3
Endrin	75.8	6.9	70.1	1.8	62.1	4.5	84.4	3.2
EPN	81.1	6.4	69.8	1.6	74.2	3.9	97.9	8.4
Epoxiconazole	79.7	6.6	70.9	2.5	71.2	3.5	99.1	7.1
EPTC	67.0	5.5	52.6	1.8	56.8	1.6	56.8	2.5
Esprocarb	77.8	9.2	71.0	3.1	70.8	2.9	81.7	3.1
Etaconazole_1	84.6	7.2	76.0	5.0	75.7	3.9	97.1	10.0
Etaconazole_2	79.1	6.3	75.8	2.4	75.5	1.4	86.1	3.7
Ethalfluralin	77.9	2.4	72.7	1.9	76.9	3.4	89.1	3.4
Ethion	82.0	7.8	71.4	2.5	72.8	6.0	82.7	4.0
Ethofumesate	87.0	9.5	70.1	2.6	73.7	4.2	94.3	3.6
Ethoprophos	77.0	6.9	70.9	3.6	73.3	3.0	85.3	5.2
Ethychlozate	83.9	11.0	72.0	4.2	68.3	4.2	77.6	7.1
Etofenprox	72.6	7.5	79.5	2.8	69.7	1.5	85.1	4.9
Etoxazole	114.2	14.7	74.9	2.6	76.5	2.0	94.5	6.6
Etridiazole	71.8	4.8	60.0	2.3	61.8	1.5	64.9	0.9
Etrimfos	75.1	8.1	73.5	3.2	75.6	1.7	87.9	2.6
Fenamidone	82.2	5.6	70.6	0.7	72.1	2.8	100.6	6.2
Fenamiphos	80.0	12.7	68.6	5.1	71.1	11.7	83.7	3.7
Fenarimol	78.8	4.6	76.0	3.7	72.4	5.0	89.2	5.8
Fenazaquin	69.3	7.1	74.3	2.6	71.8	2.3	83.4	4.1
Fenbuconazole	76.2	5.8	72.5	2.5	70.3	2.0	77.7	5.9
Fenchlorphos	75.3	5.7	71.2	2.4	70.4	0.7	81.8	4.9
Fenclorim	74.9	8.8	71.1	2.4	69.0	2.5	78.8	2.8
Fenfuram	79.8	7.2	69.5	1.8	75.0	1.8	5.6	2.3
Fenitrothion	110.3	5.5	69.1	4.5	77.7	18.6	87.2	2.7
Fenobucarb	75.9	5.9	71.1	3.3	70.9	2.2	81.5	3.2
Fenothiocarb	78.9	6.6	73.9	1.0	69.5	2.7	80.8	6.6
Fenoxanil	82.0	8.0	75.5	1.6	76.5	2.8	84.9	9.2
Fenoxycarb	57.6	9.7	70.1	1.6	71.8	12.4	53.4	2.7
Fenpropathrin	82.7	7.7	77.5	2.7	76.2	0.8	112.3	5.3
Fenson	75.5	7.2	68.8	2.7	68.7	1.2	80.6	3.7
Fenthion	80.5	7.9	67.9	2.6	72.8	2.1	83.5	6.1
Fenvalerate-1	75.1	8.1	76.6	2.9	73.7	1.1	93.4	6.6
Fenvalerate-2	83.0	8.4	77.0	4.3	68.4	3.3	99.4	8.3
Fipronil	84.1	5.2	72.3	5.5	75.7	3.1	98.3	8.7
Flamprop-isopropyl	86.1	7.1	75.8	2.9	74.2	3.3	82.1	3.9
Flonicamid	55.8	21.8	54.4	6.1	34.4	12.1	13.5	7.2
Fluazifop_butyl	81.6	9.2	73.2	2.9	73.7	2.6	68.3	14.8
Fluchloralin	77.8	3.7	65.6	3.9	74.3	5.7	89.7	2.8
Flucythrinate_1	80.9	5.6	77.1	3.8	71.3	2.4	102.7	7.1
Flucythrinate_2	81.6	5.0	76.1	3.4	74.8	1.3	101.5	7.2
Fludioxonil	83.4	11.0	71.3	4.6	72.7	2.2	113.7	7.0
Flufenpyr_ethyl	77.9	8.8	75.4	3.6	72.5	5.6	92.5	9.3
Flumetralin	86.9	6.4	74.5	2.5	70.9	9.3	73.3	10.1
Flumiclorac_pentyl	77.9	8.8	78.9	3.9	78.3	2.1	100.4	10.4
Flumioxazine	87.9	5.1	71.6	2.0	75.9	3.4	113.4	7.1
Fluopyram	82.1	10.0	75.0	3.7	74.6	2.9	89.1	8.7
Fluorodifen	84.2	8.7	62.0	2.2	71.8	9.2	80.3	2.6
Flurochloridone	80.7	8.8	68.9	3.0	69.6	3.3	95.1	4.2
Flusilazole	79.6	7.8	68.0	4.9	73.0	4.4	118.9	5.1
Flutamone	82.6	4.8	76.2	3.5	73.9	3.4	97.4	7.0
Fluthiacet_methyl	87.2	10.2	83.5	1.3	83.9	6.7	101.0	10.1
Flutianil	86.1	1.6	75.8	2.7	74.7	1.2	95.2	5.0
Flutolanil	78.6	9.1	74.4	3.7	69.6	1.2	11.7	1.7
Flutriafol	88.9	17.3	69.4	2.6	68.0	10.9	79.9	4.0
Fluvalinate-1	80.1	7.0	78.6	4.8	73.9	2.3	104.8	10.5
Fluvalinate-2	82.7	6.2	79.0	3.8	72.7	2.5	104.0	11.0
Folpet	79.8	8.5	74.1	2.8	74.1	2.6	66.5	1.1
Fonofos	76.1	6.8	70.9	3.7	72.5	1.9	83.8	4.5
Formothion	76.4	12.9	62.3	2.8	69.3	7.8	78.6	9.5
Fosthiazate-1	80.5	7.4	67.7	6.8	70.3	5.0	94.5	5.3
Fosthiazate-2	76.1	9.6	69.6	6.5	71.0	3.5	90.6	7.3
Fthalide	77.8	7.6	71.0	2.7	70.2	2.3	78.6	2.2
Furathiocarb	81.9	3.3	75.5	3.8	71.4	2.7	97.1	10.3
Halfenprox	76.0	7.4	76.1	4.4	60.6	5.7	90.8	4.6
Heptachlor	67.6	5.5	67.8	3.1	61.9	1.4	78.2	2.8
Heptachlor-epoxide	64.4	8.6	69.9	3.1	65.8	1.9	78.2	11.6
Heptenophos	76.4	6.0	72.9	3.2	72.6	2.3	77.7	2.8
Hexachlorbenzene	62.8	10.7	67.9	1.6	57.3	2.8	71.4	5.7
Hexaconazole	74.1	5.8	72.0	3.5	72.9	5.9	78.4	4.4
Imazalil	32.6	28.4	66.7	1.6	56.3	7.9	83.2	3.3
Indanofan	80.1	6.6	72.4	4.9	71.5	3.9	83.3	5.4
Indoxacarb	82.6	2.8	80.4	2.9	76.8	0.3	77.9	3.0
Iprobenfos	79.5	4.1	70.5	3.3	75.4	3.1	94.7	8.4
Iprodione	77.5	8.8	77.5	3.7	75.7	4.4	84.0	4.5
Iprovalicarb-1	80.7	8.0	74.6	3.4	71.2	5.0	82.5	5.4
Iprovalicarb-2	79.9	9.6	72.1	3.4	72.4	6.3	83.9	4.8
Isazofos	80.2	6.7	72.6	2.2	74.4	2.0	86.8	4.2
Isofenphos	82.3	6.7	72.8	3.5	74.7	2.0	86.6	4.9
Isofenphos-methyl	82.1	7.0	72.0	3.4	73.1	3.6	91.1	6.6
Isopropalin	79.1	6.4	69.8	3.2	72.9	5.0	83.0	5.5
Isoprothiolane	81.5	6.1	73.2	2.5	73.0	2.2	83.4	4.3
Isotianil	86.4	8.5	76.3	1.8	72.4	2.1	89.3	3.9
Isoxadifen-ethyl	82.4	7.8	70.8	2.7	74.2	4.2	87.4	8.1
Isoxanthion	83.9	9.5	67.0	3.1	75.3	8.9	85.3	2.3
Kresoxim-methyl	83.3	9.1	74.1	2.5	73.3	2.7	91.4	7.6
Lactofen	85.3	2.2	74.7	2.7	74.3	8.7	115.5	15.8
Leptophos	67.9	9.1	73.4	2.6	66.6	1.7	86.3	5.6
Malathion	82.0	11.3	72.5	4.2	74.8	4.7	85.6	2.3
Mecarbam	84.3	6.0	69.9	2.3	68.5	2.5	89.2	4.5
Mefenacet	77.1	7.5	75.7	3.2	73.8	2.7	91.6	4.6
Mefenpyr-diethyl	82.0	6.1	73.9	3.5	75.3	1.7	90.6	5.5
Mepronil	82.0	10.0	74.1	2.6	73.0	4.0	81.0	5.3
Metazachlor	84.6	8.0	73.2	2.9	74.7	1.7	83.3	6.9
Metconazole	78.7	5.3	73.9	1.9	73.4	2.9	95.0	9.0
Methidathion	80.6	7.5	71.0	3.3	75.9	3.7	90.7	3.8
Methoprotryne	80.4	9.4	71.2	2.9	71.8	3.9	69.8	5.7
Methoxychlor	71.9	7.6	68.1	2.3	68.7	1.6	88.1	6.1
Methyltrithion	82.3	9.3	69.7	5.1	71.5	3.3	79.0	4.2
Metolachlor	78.6	7.0	70.7	3.1	73.1	2.1	82.2	3.6
Metrafenone	74.9	7.9	75.0	4.4	73.8	0.5	87.1	7.2
Metribuzin	77.1	4.0	68.4	2.4	72.8	3.8	86.9	5.2
MGK-264_1	78.6	7.1	76.1	2.5	76.3	4.5	86.5	5.4
MGK-264_2	80.1	9.5	71.6	3.0	74.7	2.0	80.7	4.4
Mirex	53.6	9.6	70.0	1.2	48.7	3.0	77.7	4.4
Molinate	71.8	6.6	76.7	3.1	66.4	1.8	69.9	2.3
Monolinuron	69.8	2.5	68.2	2.9	72.6	3.2	83.8	4.4
Myclobutanil	79.9	9.6	71.9	2.6	70.8	4.4	97.8	9.4
Napropamide	84.5	13.3	72.6	3.7	75.1	4.7	85.3	2.2
Nitrapyrin	71.8	2.8	59.7	2.2	60.7	1.1	65.4	1.5
Nitrothal-isopropyl	85.5	3.1	65.8	3.8	74.0	6.8	89.3	6.4
Nonachlor_cis	66.3	9.8	46.4	4.8	59.7	3.2	77.3	4.8
Nonachlor_trans	61.7	11.1	47.9	4.7	58.1	2.8	81.8	3.8
Norflurazon	82.4	7.9	74.9	3.4	73.6	2.8	83.6	3.2
Nuarimol	80.0	7.8	75.9	1.9	71.4	2.7	91.1	10.1
Ofurace	77.2	7.3	73.6	1.4	73.7	4.0	87.3	5.4
Oxadiazon	79.3	8.8	75.1	3.0	75.7	1.5	96.6	6.7
Oxadixyl	56.6	22.3	61.5	6.9	26.9	8.5	80.1	3.3
Oxyflofen	86.9	3.7	72.5	2.6	73.7	11.4	67.9	14.5
Paclobutrazole	81.2	8.4	69.5	3.3	72.2	5.2	93.8	8.2
Parathion-ethyl	85.6	4.8	67.7	4.2	74.1	6.0	89.0	6.2
Parathion-methyl	79.7	2.1	70.1	3.7	76.4	3.2	87.2	2.0
Pebulate	68.6	6.1	61.7	2.0	63.7	1.5	66.2	2.2
Penconazole	83.2	8.3	70.7	3.0	73.5	2.4	83.3	1.2
Pendimethalin	80.6	5.3	67.1	4.3	76.4	5.4	90.3	7.9
Penflufen	82.3	7.1	66.4	2.9	76.3	3.1	88.3	3.9
Pentachlorobezonitrile	79.4	1.7	72.0	3.5	69.2	2.8	76.2	5.2
Penthiopyrad	81.4	8.9	76.7	2.8	75.7	4.5	1.6	0.6
Pentoxazon	78.0	5.9	74.8	2.6	74.3	1.6	87.1	5.6
Permethrin-1	71.9	8.8	80.9	3.0	71.0	2.0	83.7	4.5
Permethrin-2	130.3	8.5	77.9	2.2	67.7	4.5	86.8	5.8
Perthane	69.6	7.8	67.8	2.3	66.9	1.8	82.3	4.5
Phenothrin	97.5	7.8	81.0	4.8	66.9	3.4	96.1	13.3
Phenthoate	80.8	7.2	72.5	2.4	76.5	3.5	83.6	4.6
Phorate	75.2	5.2	63.5	1.6	70.1	3.1	79.3	5.7
Phosalone	76.7	7.5	73.3	3.7	72.3	3.3	105.9	12.1
Phosmet(PMP)	83.6	7.3	71.9	3.9	73.8	3.9	98.8	9.6
Phosphamidone	85.1	17.3	58.8	11.6	49.6	11.8	43.4	29.2
Picolinafen	80.7	6.9	72.9	3.2	76.2	1.4	91.7	5.2
Picoxystrobin	85.7	8.5	74.2	3.2	73.0	5.3	63.1	3.0
Piperonyl butoxide	79.9	8.9	76.2	2.4	74.9	2.0	102.5	8.4
Piperophos	80.9	6.0	69.4	2.3	73.8	4.1	119.1	15.3
Pirimiphos-ethyl	78.1	10.2	69.1	1.6	74.0	2.3	88.2	7.6
Pirimiphos-methyl	82.7	7.3	70.8	2.1	71.3	7.0	80.2	11.1
Pretilachlor	81.2	8.6	74.7	3.9	75.8	1.4	82.8	3.5
Primicarb	78.2	6.6	68.7	3.5	71.7	2.2	80.7	1.8
Probenazole	83.3	7.4	78.1	4.9	79.5	3.5	71.2	11.8
Prochloraz	75.9	7.0	72.6	3.5	68.6	2.2	70.7	9.8
Procymidone	83.6	8.0	75.0	2.7	75.4	2.7	87.7	4.4
Profenofos	74.1	8.6	70.1	1.3	75.9	2.8	77.3	2.6
Profluralin	81.4	1.3	71.1	3.7	74.9	3.3	88.7	7.3
Prometon	75.2	4.7	68.3	2.6	70.4	3.9	82.3	3.0
Prometryn	77.5	8.3	72.8	1.1	72.9	3.1	74.5	13.6
Pronamide	79.5	3.5	72.9	3.8	73.5	2.0	79.9	3.2
Propachlor	75.0	6.3	71.1	2.3	70.1	2.2	78.2	3.5
Propanil	83.4	7.8	73.7	2.1	76.1	4.3	82.5	6.7
Propazine	74.9	9.1	70.9	2.1	74.2	2.6	79.9	4.9
Propetamphos	77.8	4.2	72.7	4.0	72.6	3.1	93.9	6.0
Propham	72.7	14.2	66.9	5.7	57.7	3.5	75.3	12.9
Propiconazole-1	80.5	6.0	70.2	1.9	70.0	4.7	92.7	4.5
Propiconazole-2	77.8	6.7	69.9	1.5	73.6	1.5	80.4	9.0
Propisochlor	80.4	5.7	74.1	3.0	75.6	2.3	79.6	1.8
Prothiophos	76.3	7.9	72.8	2.5	66.1	3.7	84.6	4.0
Pyracabolid	85.7	8.6	71.7	2.0	72.3	3.4	86.2	6.0
Pyraclofos	72.4	6.9	73.7	4.6	75.1	6.2	98.3	7.5
Pyrazophos	77.9	3.4	74.9	5.5	75.5	6.5	110.8	10.0
Pyridaben	119.5	6.0	77.1	2.7	71.6	2.1	87.2	4.5
Pyridalyl	70.2	8.4	80.6	2.5	61.1	2.6	97.5	4.9
Pyridaphenthion	84.1	6.1	69.8	1.7	72.9	4.4	120.3	14.9
Pyrifenox 1	76.4	6.7	60.8	1.6	61.8	6.0	62.5	5.1
Pyrifenox 2	79.9	9.3	65.1	5.1	58.3	5.5	50.3	4.5
Pyrimidifen	81.7	4.2	79.7	3.3	74.7	1.3	98.8	6.2
Pyriminobac-methyl(E)	79.8	7.5	74.3	1.9	77.6	2.5	101.2	6.6
Quinalphos	82.3	5.7	69.9	2.5	69.3	3.3	86.7	7.3
Quinoxyfen	75.8	8.5	74.8	2.0	70.4	1.5	80.5	2.6
Quintozene	71.5	4.1	66.5	4.9	63.4	3.5	78.4	4.0
Sectumeton	79.8	11.9	70.3	4.2	74.2	4.1	81.2	6.3
Silafluofen	69.1	8.0	79.7	2.4	64.8	2.4	87.2	2.8
Simeconazole	76.2	5.8	70.1	2.7	71.3	3.1	88.9	7.5
Simetryn	93.6	5.5	69.0	3.7	70.6	4.5	83.1	3.8
Spiromesifen	76.6	8.7	77.6	3.3	75.4	1.3	79.8	7.5
Spiroxamine 1	68.8	4.3	68.8	3.0	68.8	3.5	81.1	8.7
Spiroxamine_2	70.2	6.4	69.6	3.1	68.8	5.2	81.5	7.3
Sulfotep	73.6	5.9	72.2	3.3	75.3	2.7	90.4	4.1
Sulprofos	76.4	7.6	75.2	2.6	70.4	3.8	84.1	4.1
TCMTB	88.1	12.7	70.7	5.9	72.0	11.5	84.1	7.1
Tebuconazole	77.7	7.4	75.6	1.9	71.0	1.8	88.1	6.8
Tebufenpyrad	79.3	4.4	74.9	2.2	75.9	1.5	86.1	5.3
Tebupirimfos	76.4	5.8	70.6	4.3	71.8	1.9	85.5	5.2
Tefluthrin	76.0	8.0	75.9	3.1	72.6	2.5	81.3	3.2
Terbacil	79.6	7.3	69.4	3.5	70.9	2.4	89.0	5.9
Terbufos	73.5	6.4	67.3	1.7	70.2	3.2	87.2	4.6
Terbumeton	74.2	9.3	69.9	2.5	71.9	2.6	82.4	6.6
Terbuthylazine	74.5	4.9	73.7	1.9	69.8	4.5	82.8	5.6
Terbutryn	77.9	8.5	74.3	2.9	73.3	3.7	89.8	2.9
Tetrachlorvinphos	76.8	5.5	72.1	3.2	70.6	3.5	86.7	2.1
Tetraconazole	81.3	9.1	69.8	4.6	73.0	5.4	89.5	4.1
Tetradifon	79.9	6.9	75.7	2.2	70.5	2.1	80.1	4.7
Tetramethrin-1	82.2	7.3	81.6	3.7	79.1	5.3	101.1	14.6
Tetramethrin-2	84.0	6.7	75.2	2.8	75.3	4.0	92.8	5.4
Tetrasul	62.7	10.9	73.2	1.7	55.5	3.2	76.0	5.4
Thiazopyr	83.2	7.0	74.7	3.1	73.2	4.4	86.8	5.8
Thifluzamide	86.1	8.6	73.9	2.5	70.8	4.8	82.8	5.8
Thiometon	71.2	4.4	49.2	9.0	68.1	3.0	60.7	4.9
Thionazin	76.5	2.6	70.0	3.5	70.8	2.4	82.5	4.7
Tolclofos_methyl	76.3	6.3	69.3	3.3	72.0	2.1	79.4	2.6
Tolfenpyrad	77.9	6.4	78.0	4.2	76.6	1.8	95.4	7.4
Tolylfluanid	81.0	7.5	76.4	2.4	71.8	3.4	71.3	5.5
Tralomethrin-1	116.4	2.6	65.3	5.4	55.1	8.8	140.2	42.3
Tralomethrin-2	78.8	2.8	77.3	5.2	73.4	3.9	91.4	3.2
Triadimefon	76.5	9.1	70.2	2.7	76.2	3.3	78.9	4.3
Triadimenol	80.0	7.9	70.7	3.4	72.6	1.9	97.8	8.5
Triallate	72.1	9.6	70.7	3.7	69.5	1.7	80.0	4.3
Triazophos	83.8	8.8	78.3	13.8	73.9	4.6	98.2	4.4
Tribufos	79.5	7.8	71.9	2.6	70.5	5.1	86.3	9.0
Tridiphane	70.7	8.3	71.6	2.7	69.0	2.4	85.4	4.6
Triflumizole	79.6	7.3	71.6	3.1	72.7	1.7	90.4	6.9
Triflumuron	82.4	4.8	75.4	2.5	72.6	0.6	84.8	4.7
Trifluralin	77.4	3.7	71.0	3.8	76.4	4.3	79.1	4.1
Uniconazole	80.5	7.5	72.5	3.6	72.1	4.8	81.4	3.7
Vernlolate	68.6	6.8	60.2	2.1	63.0	1.0	64.7	2.0
Vinclozoline	78.5	4.6	75.7	2.2	72.7	0.7	82.7	7.8
Zoxamide	95.4	11.2	74.0	3.2	70.1	1.0	110.5	32.5

**Table 5 foods-12-03001-t005:** Pesticides frequencies, concentrations, and maximum residue limits in agricultural products.

	Agricultural Product	Number of Samples	Detected Pesticides	Concentration (mg/kg)	MRL * (mg/kg)
1	Chili pepper	21	0	0	0
2	Carrot	17	0	0	0
3	Stem of garlic	7	0	0	0
4	Mango	10	Chlorpyrifos	0.02	0.4
			Fludioxonil	0.05	2.0
5	Wheat	14	0	0	0
6	Banana	24	0	0	0
7	Almond	10	0	0	0
8	Cabbage	10	0	0	0
9	Coffee bean	12	0	0	0
10	Pineapple	10	Prochloraz	0.01	5.0
				0.4	5.0
				0.02	5.0
			Fludioxonil	0.01	20
Total		135	3		

* Ministry of Food and Drug Safety (MFDS), MFDS Notification (No. 2021-26, 25 March 2021).

**Table 6 foods-12-03001-t006:** Exposure assessment of pesticides in agricultural products.

No.	Pesticide	ADI * (** mg/kg bw/day)	Agricultural Product	Concentration (mg/kg)	Mean Concentration (mg/kg)	Average Intake (g/day)	EDI *** (mg/kg bw/day)	EDI/ADI (%)
1	Chlorpyrifos	0.00016	Mango	0.02	0.0200	0.8986	3.0 × 10^−7^	0.002995
2	Fludioxonil	0.00667	Mango	0.05	0.0500	0.8986	7.5 × 10^−7^	0.000187
	Fludioxonil		Pineapple	0.01	0.0100	1.4815	2.5 × 10^−7^	0.000062
3	Prochloraz	0.00016	Pineapple	0.01	0.1433	1.4815	3.5 × 10^−6^	0.035383
	Prochloraz		Pineapple	0.4				
	Prochloraz		Pineapple	0.02				

* ADI = acceptable daily intake; ** mg/kg bw/day = mg/kg body-weight/day; *** EDI = estimated daily intake.

## Data Availability

All available data are contained within the article.

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
