# Peer review of "Simultaneous Screening of 322 Residual Pesticides in Fruits and Vegetables Using GC-MS/MS and Deterministic Health Risk Assessments"

_foods, 2023, doi:10.3390/foods12163001_

Round 1

Reviewer 1 Report

Comments and Suggestions for Authors

High throughput detection of pesticide residues in fruits and vegetables is very important for food safety risk monitoring. The author developed a GC-MS/MS method for simultaneous detection of 322 pesticide targets, which has obvious practical significance. A recoveries of 60%-120% is acceptable. However, many RSDs exceed 15%, indicating that the stability of the method is relatively poor, not mature and stable, so such a result is intolerable. Finally, the targets with RSD exceeding 15% should be removed from the 322 pesticides mentioned. In addition, the samples with RSDS of more than 15% were mostly from coffee and chilli pepper samples. Why? Please give a clear analysis and explanation in main text.

Comments on the Quality of English Language

The Quality of English Language is acceptable.

Reviewer 2 Report

Comments and Suggestions for Authors

The manuscript prepared by the authors is commendable, providing valuable insights. However, a few areas need improvement.

1. Considering the utilization of the terms "importation" and "risk" in the title, it may lead to potential ambiguity regarding the intended meaning of the financial risks associated with importing goods into the country. Hence, it is advisable to revise the title "Development of Simultaneous screening of 322 Residual Pesticides in fruits and vegetables using GC–MS/MS and their Risk Assessment for Importation" to "Development of Simultaneous Screening of 322 Residual Pesticides in Fruits and Vegetables using GC–MS/MS and Deterministic Health Risk Assessment." This modification aims to enhance clarity by emphasizing the focus on health risks while excluding explicit mention of financial risks.

2. It is recommended to orient the choice of keywords towards practicality, such as "food safety," to enhance the relevance and applicability of the study.

3. In order to adhere to a more scientific and courteous manner, I kindly request the exclusion of the word "fruit" from the specified keyword.

4. To improve the structure and comprehensiveness of the abstract, it is advisable to incorporate the following components: a clear rationale, objective, materials and methods, and conclusions. Furthermore, it is recommended to commence the abstract with a sentence that provides the rationale for conducting the study, thus setting the context for the research.

5. In the abstract, the term "322 pesticides" has been redundantly mentioned within a single sentence. It is recommended to rephrase this to enhance clarity and conciseness.

6. I kindly request clarification on the rationale underlying the selection of the matrix employed in this study.

7. In line 255, it is advised to replace the abbreviation "MMM" with "MRM".

8. I kindly request that Table 7 be revised to accurately specify the appropriate unit for both ADI and EDI as (mg/kg bw/day).

9. In order to enhance the comprehensiveness of the abstract, it is suggested to incorporate the results of the health risk assessment.

10. If you find it appropriate, I would like to suggest considering the following articles for inclusion in your research:

https://doi.org/10.1007/s11356-021-13542-0

 11. In conclusion, it would be highly beneficial to offer recommendations for future research endeavors and succinctly articulate the potential practical value of the proposed study.

Reviewer 3 Report

Comments and Suggestions for Authors

The paper presents the determination of pesticides using a multiresidue method based on GC-MS/MS. The subject is of interests, but there are several aspects that should be improved and also some experimental work should be performed in order to increase the quality of the work.

Particular comments:

·         I suppose that the chromatographic method was previously developed, but there is no reference to support it, and that in this work only its validation has been studied.

·         Line 195 “with purified extracts of the coffee, potato, corn, and red chili samples” how these extracts were obtained?. Were these extracts checked to investigate the presence of pesticides?

·         At least one chromatogram should be shown.

·         In method validation, selectivity should be assessed. Thus, authors should perform it.

·         Precision and trueness of the proposed method should be also evaluated.

·         The values of LOD and LOQ should be given

·         Other essential aspect in method validation is the evaluation of matrix effect, that in some instances and using EI it could be of significance. In this work anything is said about it. Thus, some experimental work should be performed to evaluate matrix effect.

Comments on the Quality of English Language

The quality of English language is good

Round 2

Reviewer 2 Report

Comments and Suggestions for Authors

The authors have made noteworthy revisions to the manuscript under review. Nonetheless, it is essential to address certain areas of improvement, particularly in Table 7.

Please pay attention to the attached file. Notably, the corrected units and values for the Acceptable Daily Intake (ADI) have been accurately presented in the attached file. Additionally, the revised calculations for the Estimated Daily Intake (EDI) have been appropriately depicted. It is important to acknowledge that the correct equation for obtaining the EDI in this manuscript is as follows: EDI = mean concentration (mg/kg) * mean consumption rate (kg/day) / body weight (60 kg). This clarification will enhance the accuracy and clarity of the manuscript's findings. Please apply the changes made in Table 7 to the entire text where the risk assessment results are presented.

Author Response

Thank you for your spending time. Please see the attachment.

Reviewer 3 Report

Comments and Suggestions for Authors

The authors have performed the revision process in a satisfactory way. However, there are still some aspect that should be clarified.

·         The references added to support the GC method employed should be updated.

·         Selectivity of the method should be assessed using a blank. I know this is difficult when analysing food samples, but at least a blank (without sample) covering the whole procedure (including extraction) should be performed.

·         From the authors’ answers, I understood that the matrix effect was not evaluated, but it appears in tables S1-S4. Was it evaluated?. If it was, this should be commented in the text, some compounds have a matrix effect of 40-50 %. The procedure used to evaluate the matrix effect should be described in section 2.5 ”Method validation”.

Author Response

(The authors gave the same response as above.)
